

# Atmospheric Mercury Concentrations observed at ground-based monitoring sites globally distributed in the framework of the GMOS network

Francesca Sprovieri[1], Nicola Pirrone[2], Mariantonia Bencardino[1], Francesco D'Amore[1], Francesco Carbone[1], Sergio Cinnirella[1], Valentino Mannarino[1], Matthew Landis[3], Ralf Ebinghaus[9], Andreas Weigelt[9], Ernst-Günther Brunke[8], Casper Labuschagne[8], Lynwill Martin[8], John Munthe[14], Ingvar Wängberg[14], Paulo Artaxo[18], Fernando Morais[18], Warren Cairns[15], Carlo Barbante[15], María del Carmen Diéguez[10], Patricia Elizabeth Garcia[10], Aurélien Dommergue[4,24], Helene Angot[4,24], Olivier Magand[24,4], Henrik Skov[20], Milena Horvat[17], Jože Kotnik[17], Katie Alana Read[6], Luis Mendes Neves[7], Bernd Manfred Gawlik[12], Fabrizio Sena[12], Nikolay Mashyanov[13], Vladimir Arckadievich Obolkin[23], Dennis Wip[16], Xin Bin Feng[19], Hui Zhang[19], Xuewu Fu[19], Ramesh Ramachandran[11], Daniel Cossa[22], Joël Knoery[5], Nicolas Marusczak[22], Michelle Nerentorp[21], and Claus Norstrom[20]

[1]CNR Institute of Atmospheric Pollution Research, Rende, Italy
[2]CNR Institute of Atmospheric Pollution Research, Rome, Italy
[3]US-EPA, Environmental Protection Agency, Office of Research and Development, RTP, NC,USA
[4]Univ. Grenoble Alpes, Laboratoire de Glaciologie et Géophysique de l'Environnement, Grenoble, France
[5]LBCM, Ifremer, - Centre Atlantique, Nantes, France
[6]NCAS, University of York, UK
[7]CapeVerde Observatory, INMG- São Vicente, Cape Verde
[8]Cape Point GAW Station, Climate and Environ. Research & Monitoring, South African Weather Service
[9]Helmholtz-Zentrum Geesthacht, Germany
[10]INIBIOMA-CONICET-UNComa, Bariloche, Argentina
[11]Institute for Ocean Management, Anna University, India
[12]Joint Research Centre, Italy
[13]St. Petersburg State University, Russia
[14]IVL, Swedish Environmental Research Inst. Ltd., Sweeden
[15]University Ca' Foscari of Venice, Italy
[16]University of Suriname, Dep. of Physics, Suriname
[17]Jožef Stefan Institute, Lubliana, Slovenia
[18]Univ. of Sao Paulo, Sao Paulo, Brazil
[19]State Key Laboratory of Environmental Geochemistry, Inst. of Geochemistry, Chinese Academy of Sciences
[20]AarhusUniversity, Department of Environmental Science, Denmark
[21]Chalmers University of Technology
[22]LER/PAC, Ifremer,Centre Méditerranée, La Seyne sur Mer, France
[23]Limnological Institute SB RAS, Irkutsk, Russia
[24]CNRS, Laboratoire de Glaciologie et Géophysique de l'Environnement, Grenoble, France

*Correspondence to:* Francesca Sprovieri (f.sprovieri@iia.cnr.it)

**Abstract.** Long-term monitoring data of ambient mercury (Hg) on a global scale to assess its emission, transport, atmospheric chemistry, and deposition processes is vital to understanding the impact of Hg pollution on the environment. The Global Mer-



cury Observation System (GMOS) project was funded by the European Commission (www.gmos.eu), and started in November 2010 with the overall goal to develop a coordinated global observing system to monitor Hg on a global scale, including a large network of ground-based monitoring stations, ad-hoc periodic oceanographic cruises and measurement flights in the lower and upper troposphere, as well as in the lower stratosphere. To date more than 40 ground-based monitoring sites constitute the global network covering many regions where little to no observational data were available before GMOS. This work presents atmospheric Hg concentrations recorded worldwide in the framework of the GMOS project (2010-2015), analyzing Hg measurement results in terms of temporal trends, seasonality and comparability within the network. Major findings highlighted in this paper include a clear gradient of Hg concentrations between the Northern and Southern Hemisphere, confirming that the gradient observed is mostly driven by local and regional sources, which can be anthropogenic, natural or a combination of both.

# 1   Introduction

Mercury (Hg) is found ubiquitously in the atmosphere and is known to deposit to ecosystems, where it can be taken up into food-webs and transformed to highly toxic species (i.e., methyl-Hg) which are detrimental to ecosystem and human health. A number of activities have been carried out since the late 1980s in developed countries within European and International Strategies and Programs (i.e., UNECE-CLRTAP, EU-Mercury Strategy; UNEP Governing Council) to elaborate possible mechanisms to reduce Hg emissions to the atmosphere from industrial facilities, trying to balance the increasing emissions in rapidly industrializing countries of the world (Pirrone et al., 2013); (Pirrone et al., 2008);(Pirrone et al., 2009);(Pacyna et al., 2010). Hg displays complex speciation and chemistry in the atmosphere, which influences its transport and deposition on various spatial and temporal scales (Douglas et al., 2012);(Goodsite et al., 2004);(Goodsite et al., 2012);(Lindberg et al., 2007);(Soerensen et al., 2010a);(Soerensen et al., 2010b); (Sprovieri et al., 2010b);(Slemr et al., 2015). Most of Hg is observed in the atmosphere as Gaseous Elemental Mercury (GEM/Hg0), representing 90 to 99% of the total with a terrestrial background concentration of approximately 1.5-1.7 $ng\ m^{-3}$ in the Northern Hemisphere and, between 1.0 and 1.3 $ng\ m^{-3}$ in the Southern Hemisphere based on research studies published before GMOS (Lindberg et al., 2007);(Sprovieri et al., 2010b). The results obtained from newly established GMOS ground-based sites show a background value in the Southern Hemisphere close to 1 ng m-3which is lower than that obtained in the past. Marine background concentrations show a larger variability (Soerensen et al., 2010a). Oxidized Hg species (Gaseous Oxidized Mercury or GOM) and Particulate Bound Mercury (PBM) contribute significantly to dry and wet deposition fluxes to terrestrial and aquatic receptors (Brooks et al., 2006);(Goodsite et al., 2004);(Goodsite et al., 2012)(Hedgecock et al., 2006);(Skov et al., 2006);(Gencarelli et al., 2015);(De Simone et al., 2015). Although in the past two decades a number of Hg monitoring sites have been established (in Europe, Canada, USA and Asia) as part of regional networks and/or European projects (i.e., MAMCS, MOE, MERCYMS) (Munthe et al., 2001);(Munthe et al., 2003);(Wngberg et al., 2001);(Wängberg et al., 2008);(Pirrone et al., 2003);(Steffen et al., 2008) the need to establish a global network to assess likely Southern-Northern Hemispheric gradients and long-term trends has long been considered always a high priority for policy and scientific purposes. The main reason is to make consistent and globally distributed Hg observations available





that can be used to validate regional and global scale models for assessing global patterns of Hg concentrations and deposition and re-emission fluxes. Therefore a coordinated global observational network for atmospheric Hg was established within the framework of the Global Mercury Observation System (GMOS) project (Seven Framework Program - FP7) in 2010. The aim of GMOS was to provide high-quality Hg datasets in the Northern and Southern Hemispheres for a comprehensive assessment

of atmospheric Hg concentrations and their dependence on meteorology, long-range atmospheric transport and atmospheric emissions on a global scale (Sprovieri et al., 2013). This network was developed by integrating previously established ground-based atmospheric Hg monitoring stations with newly established GMOS sites in regions of the world where atmospheric Hg observational data was scarce, particularly in the Southern Hemisphere (Sprovieri et al., 2010b). The stations are located at both high altitude and sea level locations, as well as in climatically diverse regions. The measurements from these sites have

been used to validate regional and global scale atmospheric Hg models in order to improve our understanding of global Hg transport, deposition and reemission, as well as to provide a contribution to future international policy development and implementation (Travnikov, 2016);(Gencarelli, 2016);(De Simone, 2016). Within the GMOS network, Hg measurements were carried out using high-quality techniques by harmonizing the GMOS measurement procedures with those already adopted at existing monitoring stations around the world. Standard Operating Procedures (SOPs) and QA/QC system were established

and implemented at all GMOS sites in order to assure full comparability of network observations. To ensure a fully integrated operation of the GMOS network, a centralized online system (termed the GMOS-Data Quality Management, G-DQM) was developed for the acquisition of atmospheric Hg data in near real-time and providing a harmonized QA/QC protocol. This novel system was developed for integrating data control and is based on a service-oriented approach that facilitates real-time adaptive monitoring procedures, which is essential for producing high-quality data (Cinnirella et al., 2014);(D'Amore et al.,

2015). In this paper we present for the first time a complete global dataset of Hg concentrations at selected ground-based sites in the Southern and Northern Hemispheres and highlight its potential to support the validation of global scale atmospheric models for research and policy scenario analysis.

## 2 Experimental

### 2.1 GMOS Global Network

The GMOS network currently consists of 43 globally distributed monitoring stations located both at sea level (i.e., Mace Head, Ireland; Calhau, Cape Verde; Cape Point, South Africa; Amsterdam Island, southern Indian Ocean) and high altitude locations, such as the Everest-K2 Pyramid station (Nepal) at 5050 m a.s.l. and the Mt. Walinguan (China) station at 3816 m a.s.l., as well as in climatically diverse regions, including polar areas such as Villum Research Station (VRS), Station Nord (Greenland), Pallas (Finland), and in Antarctica, Dome Concordia and Dumont d'Urville stations. It is possible to browse the

GMOS monitoring sites at the GMOS Monitoring Services web portal. The monitoring sites are classified as Master (M) and Secondary (S) with respect to the Hg measurement programs (Table 1). Master Stations perform speciated Hg measurements and collect precipitation samples for Hg analysis whereas the Secondary Stations perform only TGM/GEM measurements. Table 1 summarizes key information about GMOS stations, such as: a) the location, elevation and type of monitoring stations;





b) new sites (Master and/or Secondary) established as part of GMOS; and c) existing monitoring sites established by institutions that are part of European and International monitoring programs and managed by GMOS partners and GMOS External partners who have agreed to share their monitoring data and submit them to the central database following the interoperability principles and standards set in GEOSS (Group Earth Observation System of System). More details about the sites can also be found at:

www.gmos.eu. Eleven monitoring stations managed by external partners are included within the global network sharing their data with the GMOS central database. These new associated stations follow the "Governance and Data Policy of the Global Mercury Observation System" guidelines established by GMOS. From the start of GMOS a small number of monitoring sites have been relocated or have become recently operational, however, most of the sites have been fully operational for the entire project period, and remain active. These original core group stations consist of 27 monitoring sites. Their spatial

coverage is better throughout the Northern Hemisphere with 17 operational monitoring stations, whereas there are 5 sites in the Tropical Zone [area between the Tropic of Cancer (+23°27') and the Tropic of Capricorn(-23°27')], and 5 sites in the Southern Hemisphere. The sites in the Southern Hemisphere include new Hg stations, such as the GMOS site in Bariloche (Patagonia, Argentina), the station in Kodaicanal (South-India), and the site on the Amsterdam Island (Terres Australes et Antarctiques Françaises, TAAF) in the southern Indian Ocean, and two sites in Antarctica at the Italian-French Dome Concordia station and

at Dumont d'Urville French site.

## 3   Hg Measurements Methods

### 3.1   Field Operation

All GMOS Secondary Sites used the Tekran Continuous Mercury Vapor Analyzer, Model 2537A/B (Tekran Instruments Corp., Toronto, Ontario, Canada) with the exception of Listvyanka site (LIS), Russia and Nieuw Nikerie site (NIK), Suriname, which

used a Lumex RA-915+ mercury analyser. This last provides direct continuous GEM concentrations in air flow without Hg collection on sorbent traps (Sholupov and Ganeyev, 1995);(Sholupov et al., 2004). GMOS Master Sites used the Tekran Model 2537A/B mercury vapor analyzer coupled with their speciation system Model 1130 for GOM, and Model 1135 for PBM2.5. The principle and operation of the Tekran Hg speciation system are described in (Landis et al., 2002). Data was captured using either personal computers or data loggers and were submitted to the GMOS Central database network (www.gmos.eu/sdi).

During the implementation of the GMOS global network, harmonized Standard Operating Procedures (SOPs) as well as common Quality Assurance/Quality Control (QA/QC) protocols have been developed (Munthe et al., 2011);(Brown et al., 2010a);(Brown et al., 2010b) according to measurement practices followed within existing European and American monitoring networks and based on the most recent literature (Brown et al., 2010b);(Steffen et al., 2012);(Gay et al., 2013). The GMOS SOPs were reviewed by both GMOS partners and external partners as experts in this issue and finally adopted within GMOS

network (Munthe et al., 2011). Full SOPs are available online (www.gmos.eu/sdi) and include sections on site selection, field operations, data management, field maintenance and reporting procedures. All monitoring sites strictly followed the GMOS SOPs to harmonize operations and ensure the comparability of all results obtained worldwide. At the GMOS Master sites the Hg analyzers were operated in conjunction with the Tekran 1130/1135 speciation units, and therefore the TGM/GEM data for





**Table 1.** Characteristics of ground-based sites that are part of GMOS.

| Code | Site | Elev (m asl) | Lat | Lon | Country | GMOS Site* |
|------|------|-------------|-----|-----|---------|-----------|
| AMS | Amsterdam Island | 70 | -37,79604 | 77,55095 | Terres Australes et Antarctiques Françaises | M |
| BAR | Bariloche | 801 | -41,128728 | -71,420100 | Argentina | M |
| CAL | Calhau | 10 | 16,86402 | -24,86730 | Cape Verde | S |
| **CHE** | **Cape Hedo** | **60** | **26,86430** | **128,25141** | **Japan** | **M** |
| CPT | Cape Point | 230 | -34,353479 | 18,489830 | South Africa | S |
| CST | Celestún | 3 | 20,85838 | -90,38309 | Mexico | S |
| CMA | Col Margherita | 2545 | 46,36711 | 11,79341 | Italy | S |
| DMC | Concordia Station | 3220 | -75,10170 | 123,34895 | Antarctica | S |
| DDU | Dumont d'Urville | 40 | -66,66281 | 140,00292 | Antarctica | S |
| EVK | Ev-K2 | 5050 | 27,95861 | 86,81333 | Nepal | S |
| ISK | Iskrba | 520 | 45,56122 | 14,85805 | Slovenia | M |
| KOD | Kodaicanal | 2333 | 10,23170 | 77,46524 | India | M |
| LSM | La Seyne-sur Mer | 10 | 43,106119 | 5,885250 | France | S |
| LIS** | Listvyanka | 670 | 51,84670 | 104,89300 | Russia | S |
| LON | Longobucco | 1379 | 39,39408 | 16,61348 | Italy | M |
| MHE | Mace Head | 5 | 53,32511 | -9,90500 | Ireland | S |
| MAN | Manaus | 110 | -2,89056 | -59,96975 | Brazil | M |
| MIN | Minamata | 20 | 32,23056 | 130,40389 | Japan | M |
| MAL | Mt. Ailao | 2503 | 24,53791 | 101,03024 | China | S/M |
| **MBA** | **Mt. Bachelor** | **2743** | **43,977516** | **-121,685968** | **WA, USA** | **M** |
| MCH | Mt. Changbai | 741 | 42,40028 | 128,11250 | China | M/S |
| MWA | Mt. Walinguan | 3816 | 36,28667 | 100,89797 | China | M |
| NIK** | Nieuw Nickerie | 1 | 5,95679 | -57,03923 | Suriname | S |
| PAL | Pallas | 340 | 68,00000 | 24,23972 | Finland | S |
| RAO | Rao | 5 | 57,39384 | 11,91407 | Sweden | M |
| SIS | Sisal | 7 | 21,16356 | -90,04679 | Mexico | S |
| VRS | Villum Research Station | 30 | 81,58033 | -16,60961 | Greenland | S |

* M=Master;S= Secondary

** This sites use Lumex. Elsewhere Tekran

**In bold External GMOS partners**

these sites are explicitly referred to as GEM. GEM concentrations were also provided by the two secondary sites (LIS and NIK) which used the Lumex Hg analyser (see the Lumex measurements principle in paragraph 2.2.2). Regarding the TGM/GEM at the other GMOS Secondary sites, it has been discussed whether the Tekran 2537A/B instruments measure total gaseous Hg





(TGM = GEM + GOM) or GEM only (Slemr et al., 2011);(Slemr et al., 2015), and considering that previous modeling studies and experimental measurements highlighted that particularly at remote/background monitoring sites the oxidized fraction of the TGM is less than 2% ((Gustin et al., 2015) and references there in), we consider the Tekran 2537A/B data to represent GEM. This is also in line with studies recently published by (Slemr et al., 2015) which reports a comparison of Hg concentra-

tions at several GMOS sites in the Southern Hemisphere. Following the SOPs implemented at all GMOS sites, the Hg analyzers used at the secondary sites were operated without the speciation units but using the PTFE (Teflon) filters to protect the instrument from sea salt and other particles intrusion. (Slemr et al., 2015) assumed that the surface active GOM in the humid air of the marine boundary layer at several GMOS secondary sites, mostly located on coastline, [i.e., Cape Point (South Africa), Cape Grim (Australia) as well as Sisal (Mexico), Nieuw Nikerie (Paramaribo), Calhau (Cape Verde) etc.] has been filtered out

together with PM, partly by the sea salt particles loaded PTFE filter and partly on the walls of the inlet tubing. Consequently, they assumed that measurements at the secondary sites represent GEM only and are thus directly comparable to those at remote Master sites. On the other hand, the observations made by (Temme et al., 2003) at Troll (Antarctica) suggested that at the low temperature and humidity prevailing at this site, GOM passed the inlet tubing and the PTFE filter, measuring thus TGM and not GEM. Taking into account these findings, (Slemr et al., 2015) calculated for the GMOS Master site on Amsterdam Island

(AMS) a value of GOM less than 1% of TGM compared to the other Secondary sites in the Southern Hemisphere, including Troll, highlighting therefore a value which is insignificant when compared with the uncertainties discussed in the available peer-reviewed literature (see (Slemr et al., 2015)). Since we compare results at various stations, in this work we have taken into account analysis of both systematic and random uncertainties associated with the measurements as well as published results of Tekran intercomparison exercises as reported and discussed elsewhere ((Slemr et al., 2015) and references there in).

**3.2 GEM Measurements Method**

Amalgamation with gold is the principle method used to sample Hg0 for atmospheric measurements worldwide (Gustin, et al., 2015). The most widely used automated instrument is the Tekran 2537A/B analyser (Tekran Instrument Corp., Ontario, Canada) which performs amalgamation on dual gold cartridges used alternately, and thermal desorption (at 500°C) to provide continuous GEM measurements. One trap is sampling while the other is heated releasing Hg0 into an inert carrier gas (usually

ultra-high purity argon), quantification is by Cold Vapor Atomic Fluorescence Spectroscopy (CVAFS) at 253.7 nm (Landis et al., 2002). Concentrations are expressed in $ng\ m^{-3}$ at Standard Temperature and Pressure (STP, 273.15 K, 1013.25 hPa). The sampling interval is between 5 and 15 min based on location logistics and meteorological conditions. Taking into account the elevation of some monitoring sites in the network (i.e., Ev-K2CNR, Nepal (5050 m a.s.l.), M.Waliguan, China (3816 m a.s.l.) and Concordia Station (3220 m a.s.l.), the Tekran 2537A/B analysers have been operated with a 15-minute sample time

resolution at a flow rate of 0.8 l min-1. Following the SOPs the Tekran analysers perform also automatic internal permeation source calibrations every 71 hours, and the best estimate of the method detection limit is 0.1 $ng\ m^{-3}$ at a flow rate of 1 lmin-1. The alternative automated instrument to measure continuous GEM concentrations is the Lumex RA-915AM which is based on the use of differential atomic absorption spectrometry with direct Zeeman effect (Sholupov and Ganeyev, 1995)(Sholupov et al., 2004) providing a detection limit at sub $ng\ m^{-3}$ levels. Comparison studies between the Tekran 2537 and the RA-915AM



**Table 2.** Annually-based statistics referring to the GMOS sites for the 2013 and 2014 results

| | | \multicolumn{14}{c}{Annually- based statistics} |
|---|---|---|---|---|---|---|---|---|---|---|---|---|---|---|
| | | \multicolumn{2}{c}{Mean} | \multicolumn{2}{c}{St. Dev} | \multicolumn{2}{c}{5th} | \multicolumn{2}{c}{25th} | \multicolumn{2}{c}{50th} | \multicolumn{2}{c}{75th} | \multicolumn{2}{c}{95th} |
| | Code | 2013 | 2014 | 2013 | 2014 | 2013 | 2014 | 2013 | 2014 | 2013 | 2014 | 2013 | 2014 | 2013 | 2014 |
| Northern Hemisphere | VRS | 1,61 | 1,41 | 0,41 | 0,35 | 1,01 | 0,95 | 1,39 | 1,15 | 1,52 | 1,41 | 1,83 | 1,59 | 2,33 | 2,01 |
| | PAL | 1,45 | 1,47 | 0,11 | 0,17 | 1,27 | 1,23 | 1,38 | 1,36 | 1,46 | 1,46 | 1,53 | 1,60 | 1,61 | 1,73 |
| | RAO | 1,43 | 1,48 | 0,16 | 0,23 | 1,19 | 1,18 | 1,34 | 1,32 | 1,42 | 1,46 | 1,49 | 1,61 | 1,65 | 1,93 |
| | MHE | 1,46 | 1,41 | 0,17 | 0,14 | 1,19 | 1,17 | 1,35 | 1,34 | 1,47 | 1,41 | 1,56 | 1,49 | 1,70 | 1,61 |
| | LIS | 1,34 | 1,39 | 0,38 | 0,40 | 0,79 | 0,84 | 1,11 | 1,14 | 1,30 | 1,34 | 1,56 | 1,59 | 1,98 | 2,11 |
| | CMA | - | 1,69 | 0,36 | 0,29 | 1,40 | 1,30 | 1,59 | 1,50 | 1,88 | 1,66 | 2,11 | 1,85 | 2,54 | 2,19 |
| | MCH | 1,78 | 1,57 | 0,48 | 0,42 | 1,22 | 1,06 | 1,49 | 1,33 | 1,66 | 1,48 | 1,98 | 1,73 | 2,74 | 2,41 |
| | LON | 1,43 | - | 0,33 | - | 0,98 | - | 1,24 | - | 1,43 | - | 1,57 | - | 2,00 | - |
| | MWA | 1,33 | 1,31 | 0,64 | 0,60 | 0,63 | 0,68 | 0,90 | 0,87 | 1,33 | 1,18 | 1,59 | 1,53 | 2,65 | 2,47 |
| | MIN | 1,86 | 1,91 | 0,40 | 0,40 | 1,34 | 1,45 | 1,60 | 1,65 | 1,79 | 1,82 | 2,04 | 2,06 | 2,61 | 2,65 |
| | EVK | 1,11 | 1,33 | 0,42 | 0,22 | 0,78 | 0,98 | 0,95 | 1,18 | 1,11 | 1,32 | 1,23 | 1,46 | 1,50 | 1,70 |
| | CHE | 1,74 | 1,78 | 0,38 | 0,35 | 1,30 | 1,40 | 1,50 | 1,50 | 1,60 | 1,70 | 1,90 | 1,90 | 2,40 | 2,50 |
| | MAL | 2,04 | 1,33 | 0,64 | 0,40 | 1,32 | 0,83 | 1,60 | 1,06 | 2,04 | 1,26 | 2,33 | 1,49 | 3,26 | 2,10 |
| Tropics | SIS | 1,20 | 1,11 | 0,24 | 0,37 | 0,80 | 0,82 | 1,06 | 0,95 | 1,20 | 1,08 | 1,33 | 1,21 | 1,58 | 1,45 |
| | CAL | 1,22 | 1,20 | 0,14 | 0,09 | 1,04 | 1,08 | 1,15 | 1,14 | 1,22 | 1,19 | 1,27 | 1,25 | 1,46 | 1,36 |
| | KOD | 1,54 | 1,54 | 0,20 | 0,26 | 1,25 | 1,20 | 1,40 | 1,35 | 1,54 | 1,48 | 1,68 | 1,71 | 1,87 | 2,03 |
| | NIK | 1,13 | 1,28 | 0,42 | 0,46 | 0,43 | 0,75 | 0,81 | 1,08 | 1,12 | 1,29 | 1,43 | 1,47 | 1,78 | 1,74 |
| | MAN | 1,08 | 0.99 | 0,23 | 0,23 | 0,77 | 0,69 | 0,95 | 0,85 | 1,09 | 0,96 | 1,20 | 1,11 | 1,52 | 1,43 |
| Southern Hemisphere | AMS | 1,03 | 1,05 | 0,09 | 0,05 | 0,87 | 0,95 | 0,98 | 1,02 | 1,03 | 1,05 | 1,09 | 1,08 | 1,17 | 1,12 |
| | CPT | 1,03 | 1,09 | 0,11 | 0,12 | 0,83 | 0,89 | 0,97 | 1,01 | 1,03 | 1,09 | 1,10 | 1,18 | 1,18 | 1,27 |
| | BAR | 0,89 | 0,87 | 0,15 | 0,15 | 0,60 | 0,60 | 0,82 | 0,77 | 0,89 | 0,88 | 1,00 | 0,99 | 1,09 | 1,07 |
| | DDU | 0,85 | 0,86 | 0,19 | 0,38 | 0,57 | 0,31 | 0,75 | 0,57 | 0,85 | 0,82 | 0,93 | 1,09 | 1,15 | 1,54 |
| | DMC | 0,84 | - | 0,27 | - | 0,32 | - | 0,70 | - | 0,87 | - | 0,98 | - | 1,28 | - |

performed both during EN 15852 standard development and in the framework of the GMOS project, showed good agreement of the monitoring data obtained with these systems (Brown et al., 2010b).

## 3.3 GEM/GOM/PBM Measurements Method

Speciated atmospheric Hg measurements were performed using the Tekran Hg speciation system units (Models 1130 and 1135) coupled to a Tekran 2537A/B analyzer. PBM and GOM concentrations are expressed in picograms per cubic meter (pg m-3) at STP (273.15 K, 1013.25 hPa). At most GMOS sites, the speciation units were located on the rooftop of the station and connected to a Tekran 2537A/B analyzer through a heated PTFE line (50 °C, 10m in length). The sampling time resolution was



**Table 3.** Monthly-based statistics referring to the GMOS sites for the 2013 and 2014 results

| Region | Code | Jan 2013 | Jan 2014 | Feb 2013 | Feb 2014 | Mar 2013 | Mar 2014 | Apr 2013 | Apr 2014 | May 2013 | May 2014 | Jun 2013 | Jun 2014 | Jul 2013 | Jul 2014 | Aug 2013 | Aug 2014 | Sep 2013 | Sep 2014 | Oct 2013 | Oct 2014 | Nov 2013 | Nov 2014 | Dec 2013 | Dec 2014 |
|---|---|---|---|---|---|---|---|---|---|---|---|---|---|---|---|---|---|---|---|---|---|---|---|---|---|
| Northern Hemisphere | VRS | 1,50 | 1,45 | 1,45 | 1,47 | - | 1,55 | 1,42 | 1,13 | 1,36 | 1,64 | 1,90 | 1,48 | 1,99 | 1,48 | 1,85 | - | 1,52 | - | 1,40 | - | 1,41 | - | 1,38 | 1,11 |
| | PAL | - | 1,59 | - | 1,69 | - | 1,65 | - | 1,60 | 1,51 | 1,36 | 1,48 | 1,39 | 1,48 | 1,36 | 1,44 | 1,39 | 1,33 | 1,36 | 1,36 | 1,31 | 1,47 | 1,37 | 1,52 | 1,46 |
| | RAO | 1,46 | - | 1,50 | 1,77 | 1,37 | 1,65 | 1,34 | 1,34 | 1,30 | 1,45 | - | 1,34 | - | 1,30 | - | - | - | - | 1,27 | 1,46 | 1,31 | 1,49 | 1,46 | 1,51 |
| | MHE | - | 1,47 | 1,62 | 1,50 | 1,62 | 1,44 | 1,50 | 1,40 | 1,47 | 1,38 | 1,48 | 1,40 | 1,18 | 1,32 | 1,43 | 1,45 | 1,39 | 1,23 | 1,26 | 1,34 | 1,29 | 1,42 | 1,38 | 1,51 |
| | LIS | 1,56 | 1,45 | 1,59 | 1,50 | 1,51 | 1,53 | 1,43 | 1,41 | 1,33 | 1,35 | 1,18 | 1,32 | - | 1,33 | 1,25 | 1,34 | 1,13 | 1,24 | - | 1,36 | - | 1,40 | - | 1,56 |
| | CMA | - | - | - | - | - | 1,67 | - | 1,60 | - | 1,57 | - | 1,58 | - | 1,38 | - | 1,74 | - | 1,92 | - | 1,88 | - | 1,79 | - | 1,88 |
| | MCH | 2,03 | 1,70 | 1,70 | 1,73 | 1,91 | 1,82 | 1,72 | 1,68 | 1,77 | 1,62 | - | - | 1,37 | 1,36 | 1,91 | 1,51 | 1,51 | 1,30 | 1,74 | 1,57 | 1,92 | 1,62 | 1,78 | 1,54 |
| | LON | - | - | 1,57 | - | 1,74 | - | 1,74 | - | 1,51 | - | 1,44 | - | 1,44 | - | 1,27 | - | 1,22 | - | 1,26 | - | - | - | - | - |
| | MWA | 1,08 | 1,08 | 1,55 | 1,13 | 1,49 | 1,06 | 1,73 | 1,14 | 1,50 | 0,91 | 1,26 | - | 1,25 | - | - | 2,07 | 1,58 | 1,61 | 1,36 | 1,75 | 1,08 | 1,60 | 1,13 | 1,33 |
| | MIN | 1,92 | 1,98 | 2,01 | 1,88 | 2,00 | 1,97 | 2,04 | 1,94 | 2,25 | 2,10 | 1,63 | 1,99 | 1,56 | 1,81 | 1,58 | 1,73 | - | 1,84 | 1,85 | 1,75 | 1,86 | - | 1,82 | 1,97 |
| | EVK | 1,12 | 1,21 | 1,05 | 1,23 | 1,13 | 1,31 | 1,17 | 1,43 | 1,17 | 1,47 | 1,11 | 1,47 | 1,01 | 1,31 | - | 1,31 | 1,60 | 1,16 | - | 1,60 | - | 1,57 | - | - |
| | CHE | 1,87 | 1,78 | 1,85 | 1,79 | 1,92 | 1,93 | 1,94 | 1,91 | 1,80 | 2,08 | 1,66 | 1,99 | 1,43 | 1,74 | 1,50 | 1,48 | 1,60 | 1,57 | 1,57 | 1,41 | 1,86 | 1,57 | 1,86 | 1,74 |
| | MAL | 1,99 | 1,50 | - | 1,39 | - | 1,30 | 2,41 | 1,60 | 2,11 | 1,45 | 2,07 | 1,49 | 1,64 | - | 2,02 | - | 2,28 | - | 2,44 | - | 1,55 | 1,00 | 1,90 | 1,23 |
| | Means | **1,62** | **1,52** | **1,59** | **1,55** | **1,63** | **1,57** | **1,68** | **1,51** | **1,59** | **1,53** | **1,52** | **1,55** | **1,43** | **1,44** | **1,58** | **1,56** | **1,51** | **1,47** | **1,55** | **1,54** | **1,53** | **1,47** | **1,58** | **1,53** |
| Tropics | SIS | 1,47 | 1,04 | 1,27 | 0,96 | 1,25 | 0,96 | 1,26 | 1,20 | 1,25 | 1,22 | 1,23 | 1,17 | 1,14 | 1,06 | 1,10 | 1,17 | 1,12 | 1,12 | 1,00 | - | 1,21 | - | 1,29 | - |
| | CAL | 1,15 | 1,29 | 1,18 | 1,23 | 1,19 | 1,22 | 1,16 | 1,21 | 1,24 | - | 1,26 | 1,12 | 1,25 | 1,16 | 1,29 | 1,17 | 1,17 | 1,13 | 1,12 | 1,18 | 1,20 | 1,17 | 1,38 | 1,33 |
| | KOD | 1,61 | - | 1,57 | - | 1,62 | 1,50 | 1,65 | 1,69 | 1,63 | 1,62 | 1,39 | 1,37 | 1,39 | 1,32 | 1,44 | 1,37 | 1,35 | 1,39 | - | 1,58 | - | 1,75 | - | 1,89 |
| | NIK | - | - | - | - | 1,32 | - | 1,03 | - | 1,01 | - | 0,96 | - | 1,09 | - | - | - | 1,13 | 1,51 | 1,03 | 1,29 | 1,06 | 1,24 | 1,13 | 1,17 |
| | MAN | - | - | 1,07 | 0,98 | 1,13 | 1,02 | - | 1,10 | - | 0,99 | - | 0,96 | - | 0,94 | 1,12 | 0,95 | - | - | 1,18 | - | 1,01 | - | - | 1,05 |
| | Means | **1,41** | **1,17** | **1,33** | **1,06** | **1,30** | **1,17** | **1,27** | **1,30** | **1,28** | **1,28** | **1,15** | **1,15** | **1,22** | **1,12** | **1,24** | **1,17** | **1,19** | **1,29** | **1,08** | **1,35** | **1,12** | **1,39** | **1,26** | **1,36** |
| Southern Hemisphere | AMS | 1,03 | 1,03 | 0,98 | 1,04 | 0,98 | 1,07 | 0,98 | 1,06 | 0,89 | 1,08 | 1,08 | 1,07 | 1,12 | 1,08 | 1,12 | 1,07 | 1,05 | 1,04 | 1,00 | 1,00 | 0,99 | 0,99 | 1,10 | 1,03 |
| | CPT | 1,04 | 1,06 | - | 1,14 | 1,08 | 1,04 | 1,04 | 1,02 | 0,96 | 1,00 | 1,04 | 0,91 | 0,97 | 1,09 | 0,95 | 1,25 | 1,09 | 1,08 | 1,12 | 1,19 | 0,99 | 1,13 | 0,97 | 1,18 |
| | BAR | 0,91 | 0,85 | 0,90 | 0,74 | 0,82 | 0,79 | 0,79 | 0,89 | 0,93 | 0,92 | 0,94 | 0,89 | 0,94 | 0,95 | 0,96 | 0,95 | 0,90 | 0,92 | 0,84 | 0,92 | 0,94 | - | 0,81 | - |
| | DDU | 0,88 | 0,91 | 0,81 | 0,42 | 0,81 | - | 0,96 | 0,97 | 0,88 | 0,68 | 0,83 | - | 0,80 | 0,82 | 0,73 | - | 0,68 | - | - | - | - | 0,67 | 0,98 | 1,00 |
| | DMC | 0,69 | - | 0,68 | - | 1,16 | - | 1,16 | - | 1,01 | - | 0,93 | - | 0,89 | - | 0,75 | - | 0,85 | - | 0,75 | - | 0,66 | - | 0,84 | - |
| | Means | **0,91** | **0,96** | **0,84** | **0,84** | **0,97** | **0,97** | **0,99** | **0,99** | **0,93** | **0,92** | **0,96** | **0,96** | **0,94** | **0,98** | **0,90** | **1,09** | **0,92** | **1,02** | **0,93** | **1,04** | **0,90** | **0,93** | **0,94** | **1,07** |





5 min for GEM and 2hrs for GOM and PBM at most of the GMOS stations, with sampling flow rate of $10\ l\ min^{-1}$. Speciation measurements were performed following the GMOS SOPs and procedure as described elsewhere (Landis et al., 2002) using a size selective impactor inlet (2.5 $\mu m$ cut-off aerodynamic diameter at $10\ l\ min^{-1}$), a KCl-coated quartz annular denuder in the 1130 unit, and a quartz regenerable particulate filter (RPF) in the 1135 unit.

## 3.4 Quality Assurance and Quality Control Procedures

In terms of network data acquisition, QA/QC implementation procedures, and data management, the worldwide configuration of the GMOS network was a challenge for all scientists and site operators involved in GMOS. The traditional approaches to Hg monitoring QA/QC management that were primarily site specific and manually implemented, were no longer easily applicable or sustainable when applied to a global network with the number and size of data streams generated from the monitoring stations in near real- time. The G-DQM system was designed to automate the QA process making it available on the web with a user-friendly interface to manage all the QC steps from initial data transmission through final expert validation. From the user's point of view, G-DQM is a web-based application, developed using a software as a Service (SaaS)-based approach (D'Amore et al., 2015). G-DQM is part of the GMOS Cyber-Infrastructure (CI), which is a research environment that supports advanced data acquisition, storage, management, integration, mining and visualization, built on an IT infrastructure (Cinnirella et al., 2014);(D'Amore et al., 2015).

## 4 Results and Discussion

### 4.1 GMOS Data Coverage and Consistency

Almost all GMOS stations provide near real-time raw data that are archived and managed by GMOS-CI. Figure 1 and 2, over the 2011-2015 period, and at some of the ongoing Secondary and Master GMOS Stations, show the elemental and speciated Hg row data coverage, respectively. During the first year of the project a number of sites were being established and/or equipped and not enough data was available to support broad network spatial analysis. Most GMOS sites started their measurements at the end of 2011, therefore in the present paper we will refer the discussion mainly to the 2013 and 2014 Hg data both for GEM and GOM and PBM concentrations and their trends.

### 4.2 Northern - Southern Hemispheric Gradients

A summary of descriptive statistics based on monthly and annual averages from all GMOS sites is presented in Tables 2 and 3. The 2013 and 2014 annual mean concentrations of 1.55 and 1.51 $ng\ m^{-3}$, respectively for the sites located in the Northern Hemisphere were calculated by averaging the 13 site medians for both years. Similar calculations were made for the Southern Hemisphere and the Tropics (see Table 2 and 3). Annual median concentrations of 1.23 and 1.22 $ng\ m^{-3}$ for 2013 and 2014, respectively were obtained in the Tropical zone, and 0.93 and 0.97 $ng\ m^{-3}$ for the Southern Hemisphere. Figure 3 shows the GEM yearly distribution for 2013 (blue) and 2014 (green). The sites have been organized in the graphic as well as in the




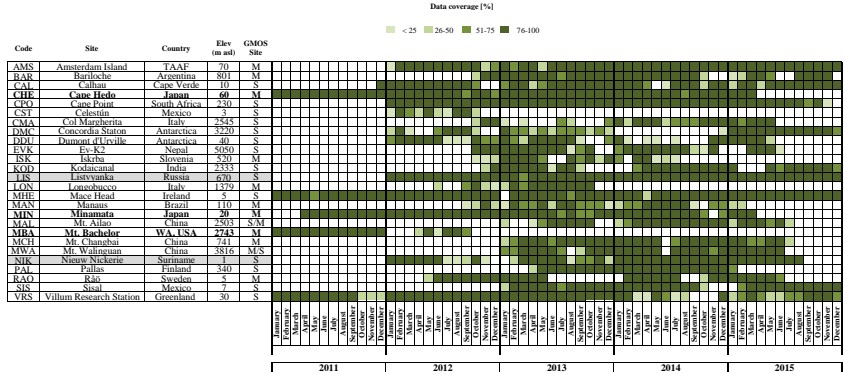

**Figure 1.** Coverage and consistency (%), on monthly basis, of GEM data collected at some of the on-going GMOS Secondary stations, over the period 2011-2015

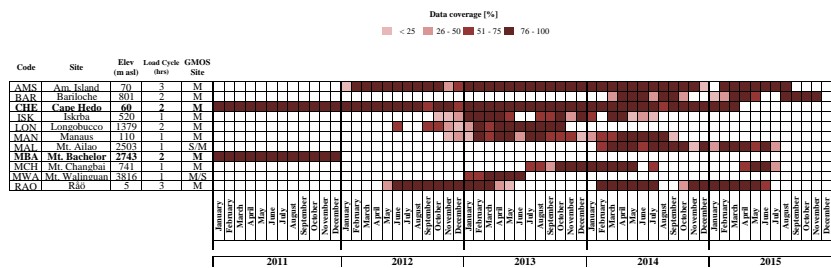

**Figure 2.** Coverage and consistency, on monthly basis, of GOM/PBM data collected at some of the on-going GMOS Master stations, over the period 2011-2015

Tables according to their location in the Northern Hemisphere, in the Tropics and those in the Southern Hemisphere. The data so far does not cover a long enough time-span to investigate temporal trends, however some attempts have been previously made for the more established sites, such as Mace Head (MHE), Ireland (Ebinghaus et al., 2011)(Weigelt et al., 2015), and Cape Point (CPT), South Africa (Slemr et al., 2015). At MHE the annual baseline TGM means observed by (Ebinghaus et al.,

5 2011) decreased from 1.82 $ng\ m^{-3}$ at the start of the record in 1996 to 1.4 $ng\ m^{-3}$ in 2011 showing a downwards trend of 1.4-1.8% per year. Both a downward trend of 1.6% at MHE from 2013 and 2014 and the slight increase in Hg concentrations seen by (Slemr et al., 2015) at CPT from 2007 to 2013 continued throughout the end of 2014. Some debate remains as whether anthropogenic emissions are increasing or decreasing ((Lindberg et al., 2002);(Selin et al., 2008);(Pirrone et al., 2013) and references therein). A clear gradient of GEM concentrations between the Northern and Southern Hemispheres is seen in the

10 data for both 2013 and 2014, in line with previous studies (Soerensen et al., 2010a);(Soerensen et al., 2010b);(Sommar et al., 2010);(Lindberg et al., 2007);(Sprovieri et al., 2010b).





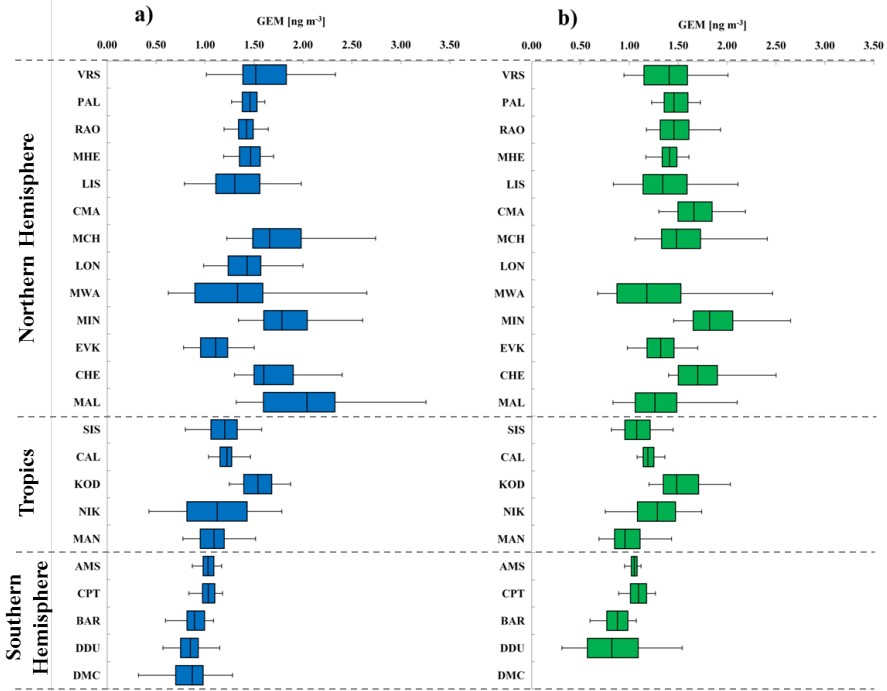

**Figure 3.** Box-and-whisker plots of gaseous elemental mercury yearly distribution (GEM, $ng\ m^{-3}$) at all GMOS stations for a) 2013 and b) 2014 years. The sites are organized according to their latitude from the Northern to the southern locations. Each box includes the median (midline), 25th and 75th percentiles (box edges), 5th and 95th percentiles (whiskers).

The 13 northern sites had significantly higher median concentrations than did the southern sites. The north-south gradient is clearly evident in Figure 4 and 5 where are reported, respectively, the probability density functions (PDFs) of the raw data with a sampling time $\Delta_t = 300$ sec, and the monthly averaged data. The datasets have been divided into three principal groups related to the longitude: north samples, tropical samples and south samples. A small overlap can be seen in the three distributions, and

5    the experimental data (dash dotted lines in figures 4 and 5) can be fitted trough a Log-normal distribution (full line in figures 4 and 5). In order to make clear the distinction between the distributions we perform the standard t-Student test against the null hypothesis ($h_0$) that the three distribution comes from the same mother distribution with the same mean ($\mu_0$) and unknown standard deviation ($\sigma_0$).

For every case the null hypothesis ($h_0$) can be rejected, say the means of the three distribution are significantly different, with

10    a 99% confidence level. If XN, XS and XT are the mean of the experimental measures respectively for the Northern, Southern and Tropical groups, the confidence intervals evaluated from the t-Student test between these values are reported in Table 4. The interpretations of the results clearly demonstrate that XN > XS > XT (Table 4), so that there exist a significant gradient in the GEM concentrations from Northern Hemisphere to the Southern Hemisphere. Due to the significant difference in the PDFs, the probability p (p-value) of observing a test statistic as extreme as, or more extreme than, the observed value under



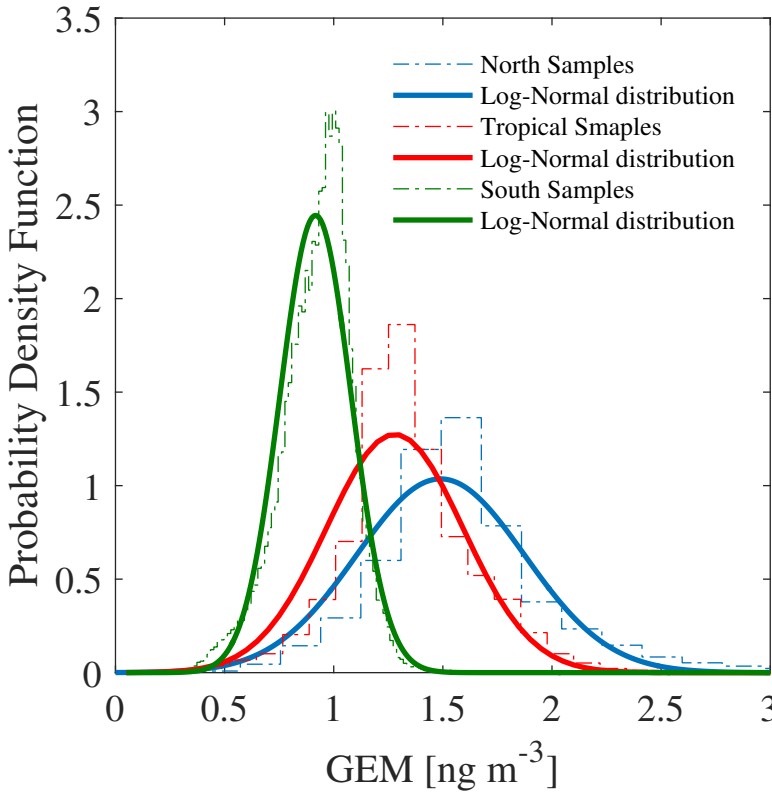

**Figure 4.** Probability density functions (PDFs) of the GEM raw data ($ng\,m^{-3}$) with a sampling time $\Delta_t = 300$ sec for the Northern, Southern and Tropical samples group (dash dotted lines). Full lines the Log-Normal distribution fit of the samples.

**Table 4.** The mean (X) of the experimental measures respectively for the Northern ($X_N$), Southern ($X_S$), and Tropical ($X_T$) groups, and the confidence intervals evaluated from the t-Student test among them

| Difference between means | Minimum of the confidence interval | Maximum of the confidence interval |
|:---:|:---:|:---:|
| $X_N - X_S$ | 0.5896 | 0.592 |
| $X_N - X_T$ | 0.225 | 0.2287 |
| $X_T - X_S$ | 0.362 | 0.365 |

the null hypothesis is close to zero. So that the validity of the null hypothesis should be rejected. The spatial gradient observed from north to south regions is also highlighted in both Figures 6 and 7 that also report the statistical monthly distribution of



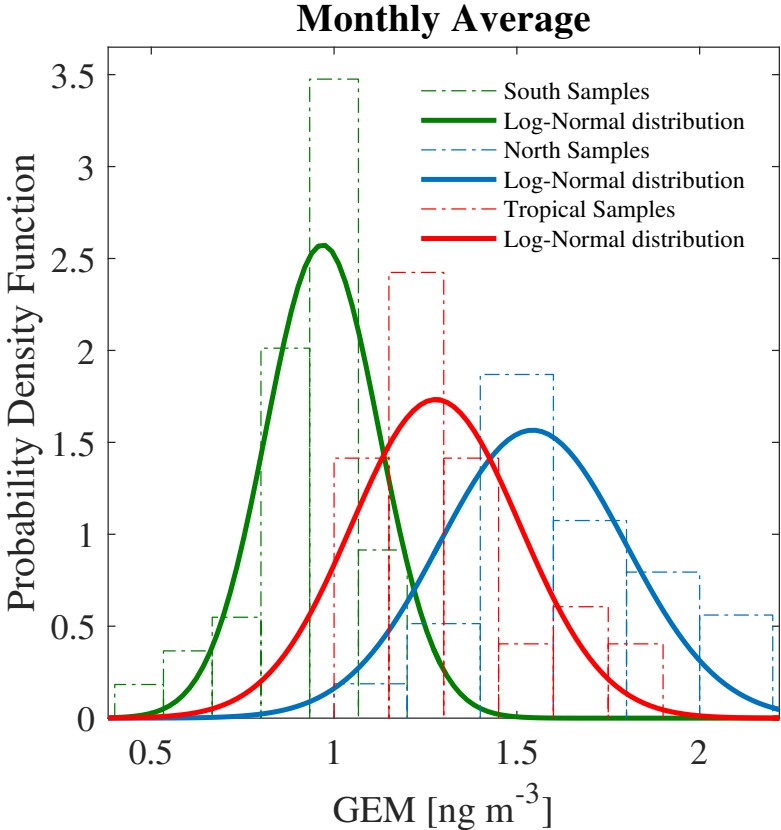

**Figure 5.** Probability density functions (PDFs) of the Monthly Averaged GEM data ($ng\ m^{-3}$) for the Northern, Southern and Tropical samples group (dash dotted lines). Full lines the Log-Normal distribution fit of the samples.

**Table 5.** The mean (X) of the experimental measures respectively for MCH, MWA, and MAL, confidence intervals and p-value associated to the t-Student test among them

| Difference between means | Minimum of the confidence interval | Maximum of the confidence interval | P-value |
|---|---|---|---|
| $X_{MCH} - X_{MWA}$ | 0.130 | 0.135 | 0.43 |
| $X_{MCH} - X_{MAL}$ | 0.233 | 0.237 | 0.19 |
| $X_{MWA} - X_{MAL}$ | 0.0.365 | 0.369 | 0.08 |

GEM values obtained for 2013 and 2014, respectively at all GMOS sites in the Northern and Southern Hemispheres as well as in the Tropical area.





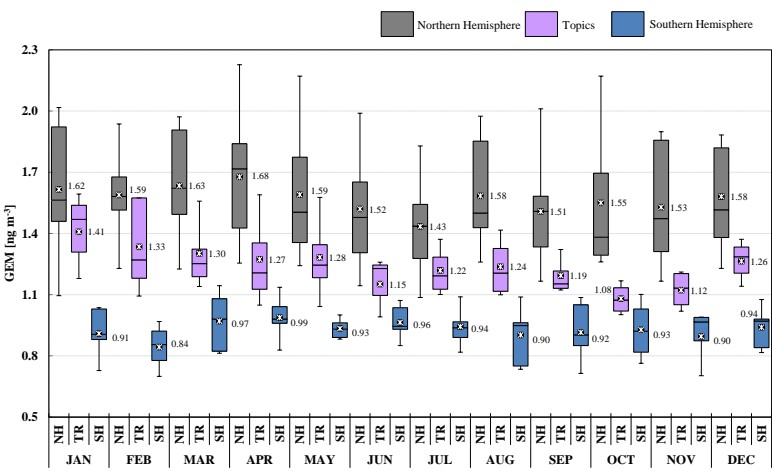

**Figure 6.** Monthly statistical distribution and spatial gradient for 2013 yr from Northern Hemisphere to Southern Hemisphere

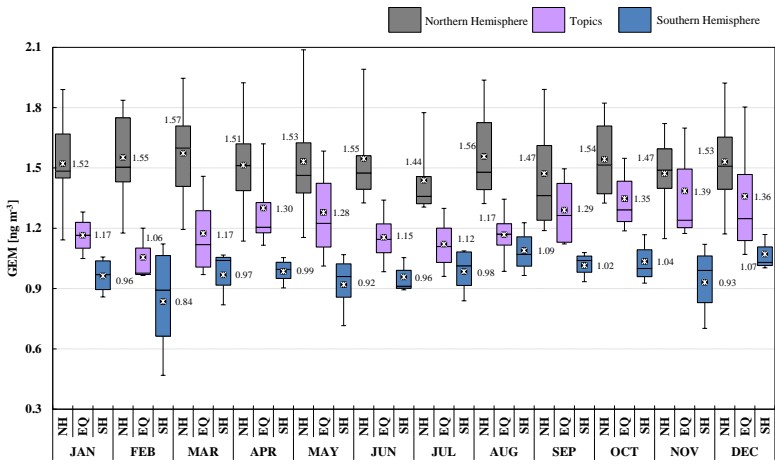

**Figure 7.** Monthly statistical distribution and spatial gradient for 2014 from Northern Hemisphere to Southern Hemisphere



### 4.2.1 Seasonal Patterns analysis in the Northern Hemisphere

Statistics describing the spatial and temporal distribution of GEM concentrations at all GMOS sites for 2013 and 2014 are summarized in Figure 3 whereas Figures 6 and 7 show the monthly statistical GEM distribution for both years considered. The GEM concentrations highlight that the median and mean GEM values of most of the GMOS sites were between 1.3 and 1.6

$ng\ m^{-3}$, with a typical interquartile range of about 0.25 $ng\ m^{-3}$. Only a few sites have shown a median and mean values above 1.6 $ng\ m^{-3}$, such as MCH, MIN, and MAL, and only the EVK site, located at 5050 m a.s.l. in the eastern Himalaya Mountains of Nepal, reported median and mean values below 1.3 ng m-3. This value is comparable with free tropospheric concentrations measured in August 2013 over Europe (Weigelt et al., 2016). The mean GEM concentration observed at EVK is less than the reported background GEM concentration for the northern hemisphere (1.5-1.7 ng m-3) and more similar to

expected background levels of GEM in the Southern Hemisphere (1.1-1.3 $ng\ m^{-3}$) (Lindberg et al., 2007);(Pirrone, 2016). The values between 1.3 and 1.6 $ng\ m^{-3}$ observed at the other GMOS sites in the Northern Hemisphere are comparable to the concentrations measured at the long-term monitoring stations at Mace Head, Ireland (Ebinghaus et al., 2011);Slemr et al. (2011);Weigelt et al. (2015) and Zingst, Germany Kock et al. (2005). GEM concentration means are also in good agreement with the overall mean concentrations observed at multiple sites in the Canadian Atmospheric Mercury Measurement Network

(CAMNet) (1.58 ng m-3) reported by (Temme et al., 2007) and those reported from Arctic stations in this paper (STN, PAL). Seasonal variations of GEM concentrations have been also observed at all GMOS sites in the Northern Hemisphere. Most sites show higher concentrations during the winter and spring, and lower concentrations in summer and fall seasons (Figures 6 and 7). However, few sites such as VRS, Station Nord (north-eastern Greenland, 81°36' N, 16°40'W) show a slightly different seasonal variation. In winter this High Arctic site (VRS) is sporadically impacted by episodic transport of pollution

mainly due to high atmospheric pressure systems over Siberia and low pressure systems over the North Atlantic (Skov et al., 2004);(Nguyen et al., 2013). During the spring (April-May) and summer (August-September) seasons GEM concentrations show a higher variability with low concentrations near the instrumental detection limit due to episodic atmospheric Hg depletion events (AMDEs) that occur in the Spring (Skov et al., 2004);(Sprovieri et al., 2005a);(Sprovieri et al., 2005b);(Hedgecock et al., 2008);(Steffen et al., 2008);(Dommergue et al., 2010a), and high GEM concentrations ( 2 $ng\ m^{-3}$) in June and July,

probably due to GEM emissions from snow and ice surfaces (Poulain et al., 2004);(Sprovieri et al., 2005a);(Sprovieri et al., 2005b);(Sprovieri et al., 2010b); (Dommergue et al., 2010b);(Douglas et al., 2012) and Hg evasion form the Arctic Ocean (Fisher et al., 2012);(Dastoor and Durnford, 2014). Models of the Marine Boundary Layer (MBL) that simulate the temporal variations of Hg species (Hedgecock and Pirrone, 2005);(Hedgecock and Pirrone, 2004);(Holmes et al., 2009);(Soerensen et al., 2010b) show that photo-induced oxidation of GEM by Br can reproduce the diurnal variation of GOM observed in the MBL

during cruise measurements better than other oxidation candidates (Hedgecock and Pirrone, 2005);(Sprovieri et al., 2010a) and also the seasonal variation (Soerensen et al., 2010b). Although Br is currently considered to be the globally most important oxidant for determining the lifetime of GEM in the atmosphere, there are also other possible candidates that can enhance Hg oxidation (Hynes et al., 2009);(Ariya et al., 2008);(Subir et al., 2011);(Subir et al., 2012). The lack of a full understanding of the reaction kinetics and fate of atmospheric Hg highlights the need to have a global observation system as presented here in





### 4.2.2 GMOS Sites in Asia

As can be seen in Figure 3, the group with the highest GEM median variability and maximum concentrations is in Asia which
include the following sites: Mt. Ailao (MAL), Mt. Changbai (MCH), Mt. Waliguan (MWA) and Minamata (MIN), where 95th
percentile values ranged from 3.26 to 2.74 $ng\,m^{-3}$ in 2013 (Table 2). These sites are often impacted by air masses that have
crossed emission source regions (AMAP/UNEP, 2013). GEM concentrations recorded at all remote Chinese sites (MAL, MCH,
and MWA) are elevated compared to that observed at background/remote areas in Europe and North America, and at others
sites in the Northern Hemisphere ((Fu et al., 2012a);(Fu et al., 2012b);(Fu et al., 2015) and references therein). A previous study
by (Fu et al., 2012a) at MWA suggested that long-range atmospheric transport of GEM from industrial and urbanized areas in
north-western China and north-western India contributed significantly to the elevated GEM at MWA. MAL station is located
in South-western China, at the summit of Ailao Mountain National Nature Reserve, in central Yunnan province. It is a remote
station, isolated from industrial sources and populated regions in China. Kunming, one of the largest cities in South-western
China, is located 180 km to the northeast of the MAL site. The winds are dominated by the Indian summer monsoon (ISM) in
warm seasons (May to October), and the site is mainly impacted by Hg emission from eastern Yunnan, western Guizhou, and
southern Sichuan of China and the northern part of the Indochinese Peninsula. In cold seasons the impact of emissions from
India and north-western part of the Indochinese Peninsula increased and played an important role in elevated GEM observed
at MAL (Zhang et al., 2015). However, most of the important Chinese anthropogenic sources of Hg and other air pollutants
are located to the north and east of the station, whereas anthropogenic emissions from southern and western Yunnan province
are fairly low (Wu et al., 2006);(Kurokawa et al., 2013);(Zhang et al., 2015). Average atmospheric GEM concentrations during
this study calculated for MWA and MAL during 2013 and 2014 are in good agreement with those observed during previous
measurements at both sites from October 2007 to September 2009 at MWA and from September 2011 to March 2013 at
MAL (Fu et al., 2015);(Zhang et al., 2015). Also the overall mean GEM concentration observed in 2013 and 2014 at MCH
background air pollution site (1.66±0.48 $ng\,m^{-3}$, in 2013 and 1.48±0.42 $ng\,m^{-3}$, in 2014, respectively), is in good agreement
with the overall mean value recorded earlier from 24 October 2008 to 31 October 2010 (1.60±0.51 $ng\,m^{-3}$, Fu et al., 2012).(Fu
et al., 2012a) highlighted higher mean TGM concentration of 3.58±1.78 $ng\,m^{-3}$ observed from August 2005 to July 2006,
probably due to surface winds circulation with effect of regional emission sources, such as large iron mining district in Northern
part of North Korea and two large power plants and urban areas to the southwest of the sampling site. In summary, the observed
concentrations are a function of site location relative to both natural and anthropogenic sources, elevation, and local conditions
(i.e., meteorological parameters), often showing links to the patterns of regional air movements and long-range transport.
Seasonal variations at ground-based remote sites in China have been observed. At MCH GEM was significantly higher during
cold seasons compared to that recorded in warm seasons (from April to September) whereas the reverse has been observed
at the other two Chinese GMOS sites. In order to statistically check the difference of GEM concentrations among the three
Chinese sites the statistical test has been performed. In particular, Figure 8 reports the probability density function (PDF)



for the three Chinese sites, MCH, MWA, and MAL. The graphic shows a large overlap in the data distribution of MCH and MAL, whereas the third Chinese site, MWA is centered on lower values. Since the two distributions are similar, the standard t-Student test has been carried out against the null hypothesis ($h_0$) that the three distribution comes from the same mother distribution with the same mean ($\mu_0$) and unknown standard deviation ($\sigma_0$). For every case the null hypothesis ($h_0$) can be

rejected, say the means of the three distribution are significantly different, with a 99% confidence level. The mean (X) of the experimental measures respectively for MCH (XMCH), MWA (XMWA), and MAL (XMAL), the confidence intervals, as well as the associated probability (p-value), evaluated from the t-Student test among these values are reported in Table 5. Due to the significant difference in the PDFs, between MWA and MAL sites, and between MCH and MAL, the probability p (p-value) of observing a test statistic as extreme as, or more extreme than, the observed value under the null hypothesis is not significant.

The singular case is represented for MCH – MWA, however the p-value is lower than 0.5, so the validity of the null hypothesis should be rejected also in this case.

Several hypothesis have been made to explain the seasonal variations of GEM in China, including seasonal changes in anthropogenic GEM emissions and natural emissions. The seasonal emission changes mainly resulted from coal combustion for urban and residential heating during cold seasons. This source lacks emission control devices and releases large amounts

of Hg leading to elevated GEM concentrations in the area, and thus at MCH (Feng et al., 2004);(Fu et al., 2008a);(Fu et al., 2008b);(Fu et al., 2010). Conversely, GEM at MAL and MWA was higher in warm seasons than in cold seasons. These findings highlight that emissions from domestic heating during the winter could not explain the lower winter GEM concentrations observed at MWA and MAL but there might be other not-yet-understood factors that played a key role in the observed GEM seasonal variations at these sites, such as the monsoonal winds influence which can change the source–receptor relationship at

observational sites and subsequently the seasonal GEM trends (An, 2000);(Fu et al., 2015). Among the remote Chinese sites, MAL started as Secondary site and in 2014 was upgraded to a Master site; conversely, MWA started as Master site and then became a Secondary site whereas MCH operated continuously as Master site. Therefore, PBM and GOM concentrations have been measured during the 2013 and 2014 at all Chinese sites even if not continuously (see Figure 2 for Hg speciation data coverage). The GEM and PBM concentrations measured at these sites were substantially elevated compared to the background

values in the Northern Hemisphere, from 1.8 to 42.8 $pg\ m^{-3}$ and from 40.4 to 167.4 $pg\ m^{-3}$ at the MCH and MWA respectively, in 2013. The 2014 PBM maxima were 44.2 and 45.0 $pg\ m^{-3}$ at MCH and MAL, respectively. Regional anthropogenic emissions and long-range transport from domestic source regions are likely to be the primary causes of these elevated values (Sheu et al., 2013). Seasonal variations of PBM observed at the Chinese Master sites mostly showed lower concentrations in summer and higher concentrations (up to 1 order of magnitude higher) in winter and fall (Wang et al., 2006);(Wang et al.,

2007);(Fu et al., 2008b);(Zhu et al., 2014);(Xu et al., 2015);(Xiu et al., 2009);(Zhang et al., 2013). The higher PBM in winter was likely caused by direct PBM emissions, formation of secondary particulate Hg via gas-particle partitioning and a lack of wet scavenging processes (Z. W. Wang et al., 2006; Fu et al., 2008b; Zhu et al., 2014). PBM has an atmospheric residence time ranging from a few hours to several days and can therefore be transported to the remote sites when conditions are favourable (Sheu et al., 2013). Atmospheric particulate matter (PM) pollution is of special concern in China due to the spatial distribution

of anthropogenic emissions concentrations of PM2.5 in heavily populated areas of eastern and northern China are among the



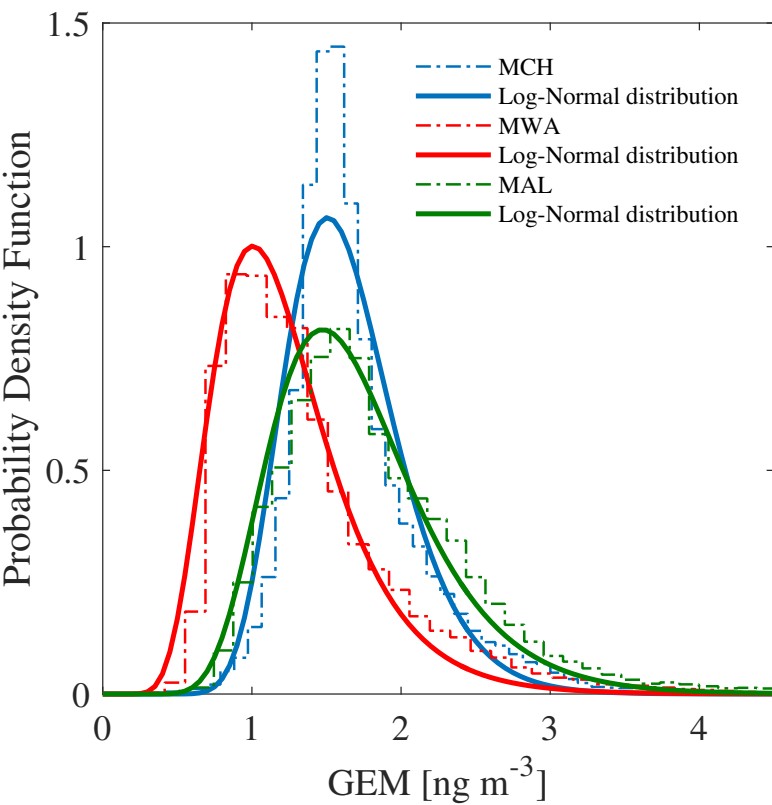

**Figure 8.** Probability density functions (PDFs) of the GEM raw data ($ng\ m^{-3}$) with a sampling time $\Delta_t$ = 300 sec for the Chinese samples group (dash dotted lines). Full lines the Log-Normal distribution fit of the samples.

highest in the world (van Donkelaar et al., 2010). The GOM concentrations observed at both master sites show high variability and several episodes with high GOM values were probably due to local emission sources (such as domestic heating in small settlements) rather than to long-range transport from industrial and urbanized areas (Fu et al., 2015). GOM has a shorter atmospheric residence time that limits long-range transport (Lindberg and Stratton, 1998);(Pirrone et al., 2008). However, with low

5  RH and high winds, the possibility of regional transport of GOM cannot be ruled out. For example, the observations at MWA exhibit a number of high-GOM events related to air plumes originating from industrial and urbanized centres that are about 90 km east of the sampling site (Fu et al., 2012a);(Pirrone, 2016). MWA is a remote site situated at the edge of north-eastern part of the Qinghai–Xizang (Tibet) Plateau. The monitoring station is relatively isolated from industrial point sources and there are no known local Hg source around the site. Most of the Chinese industrial and populated regions associated with anthropogenic

10  Hg emissions are situated to the east of MWA. Predominantly winds are from the west to southwest in cold seasons and the east





in warm seasons (Pirrone, 2016). East Asia is, in fact, the largest Hg source region in the world, contributing to nearly 50% of the global anthropogenic Hg emissions to the atmosphere (Streets et al., 2005);(Streets et al., 2011);(Pirrone et al., 2010);(Lin et al., 2010).

### 4.2.3 Seasonal Patterns analysis in the Southern Hemisphere

For the sites located in the Southern Hemisphere, the GEM concentrations highlight that the median and mean GEM values ranged between 1.1 and 1.5 $ng\ m^{-3}$, in both 2013 and 2014, with a typical interquartile range of about 0.25 $ng\ m^{-3}$ (see Figures 3, 6 and 7). The mean GEM concentrations observed at the southern sites are lower than those reported in the Northern Hemisphere but in good agreement with the southern hemispherical background (1.1 $ng\ m^{-3}$) (Lindberg et al., 2007);(Sprovieri et al., 2010b);(Lindberg et al., 2002);(Dommergue et al., 2010b);(Angot et al., 2014);(Slemr et al., 2015);(So-

erensen et al., 2010a), and the expected range for remote sites in the Southern Hemisphere. As in the Northern Hemisphere, a seasonal variation of GEM concentrations was observed in the Southern Hemisphere. In particular, GEM concentrations from the coastal Global Atmosphere Watch station, Cape Point (CPT), South Africa show seasonal variations with maxima during austral winter and minima in summer. The site is located in a nature reserve at the southern-most tip of the Cape Peninsula on a hill, 230 m a.s.l. It is characterized by dry summers with moderate temperatures and increased precipitation (cold fronts)

during austral winter. During the summer months, biomass burning events sometimes occur within the south-western Cape region affecting GEM levels. The dominant wind direction at CPT is from the southeastern sector advecting clean, maritime air from the South Atlantic Ocean (Brunke et al., 2004);(Brunke et al., 2012) which occur primarily during austral summer (December till February). Furthermore, the station is also at times subjected to air from the northern sector, mainly during austral winter. During such continental airflow events, anthropogenic emissions from the industrialized area in Gauteng, 1500 km to

the north-east of CPT, can sometimes be observed. The GEM seasonal variability at CPT is hence in good agreement with the prevailing climatology at the site. Also GEM data at Amsterdam Island followed a similar trend, with slightly but significantly higher concentrations in winter (July–September) than in summer (December–February). Amsterdam Island is a remote and very small island of 55 $km^2$ with a population of about 30 residents, located in the southern Indian Ocean at 3400 km and 5000 km downwind from the nearest lands, Madagascar and South Africa, respectively (Angot et al., 2014). GEM concentrations at

AMS were remarkably steady with an average hourly mean concentration of 1.03±0.08 $ng\ m^{-3}$ and a range of 0.72–1.55 $ng\ m^{-3}$. A small seasonal cycle has been observed by (Angot et al., 2014) and despite the remoteness of the island, wind sector analysis, air mass back trajectories and satellite observations suggest the presence of a long-range contribution from the southern African continent to the GEM regional/global budget from July to September during the biomass burning season extended from May to October (Angot et al., 2014). The higher GEM concentrations at AMS are comparable with those recorded at

Calhau (Cape Verde), Nieuw Nickerie (Paramaribo), and Sisal (Mexico) in the Tropical zone, whereas the lower concentrations of GEM observed, less than 1 $ng\ m^{-3}$, were associated with air masses coming from southern Indian Ocean and the Antarctic continent. Bariloche (BAR) Master site in North Patagonia also shows higher concentrations during the austral winter (from end of May to September), and lower concentrations in other seasons (Diéguez et al., 2015). The Patagonian site has been established inside Nahuel Huapi National Park, a well-protected natural reserve, located at the east of the Patagonian Andes.





The area is included in the Southern Volcanic Zone (SVZ) of the Andes, under the influence of at least three active volcanoes with high eruption frequency located at the west of the Andes cordillera, (Daga et al., 2014). The climate of the region is influenced by the year-round strong westerly winds blowing from the Pacific which discharge the humidity in a markedly seasonal way (fall-winter) in the western area of the Park. GEM records at BAR station show background concentrations comparable

to that found in Antarctica and other remote locations of the South Hemisphere with concentrations ranging between 0.2 and 1.3 $ng\ m^{-3}$, with an annual mean of $0.89 \pm 0.15\ ng\ m^{-3}$. Previous records of GEM concentrations from a short-term survey in 2007 along a longitudinal transect across the Andes with Bariloche as the eastern endpoint, reported concentrations below 2 $ng\ m^{-3}$ close to BAR. In this survey, the highest GEM concentrations were recorded in the proximity and downwind from the volcanic area reaching concentrations up to 10 $ng\ m^{-3}$ (Higueras et al., 2014). Similarly to the seasonal trends at other

GMOS sites in the Southern Hemisphere, GEM concentrations were at their lowest level in summer on the Antarctic Plateau at Concordia Station (DMC, altitude 3220 m) but at their highest level in fall (Angot et al., 2016b). GEM concentrations reached levels of 1.2 $ng\ m^{-3}$ from mid-February to May (fall) likely due to a low boundary layer oxidative capacity under low solar radiation limiting GEM oxidation, and/or a shallow boundary layer ($\sim$ 50 m in average) limiting the dilution. In summer (November to mid-February), the DMC GEM data showed a high variability with a concentration range varying from below

the detection limit to levels comparable to those recorded at mid-latitude background Southern Hemisphere stations due to an intense chemical exchange at the air/snow interface. Additionally, the mean summertime GEM concentration at DMC was $\sim$ 25% lower than at other Antarctic stations at the same period of the year, suggesting a continuous oxidation of GEM as a result of the high oxidative capacity of the Antarctic plateau boundary layer in summer. GEM depletion events occurred each year in summer (January-February 2012 and 2013) with GEM concentrations remaining low ($\sim$ 0.40 $ng\ m^{-3}$) for several

weeks. These depletion events did not resemble to the ones observed in the Arctic. They were not associated with depletion of ozone and occurred as air masses stagnated over the Plateau which could favor an accumulation of oxidants within the shallow boundary layer. These observations suggest that the inland atmospheric reservoir in Antarctica is depleted in GEM and enriched in GOM in summer. Measurements at DDU on the East Antarctic coast (Angot et al., 2016a) were dramatically influenced by air masses exported from the Antarctic Plateau by strong katabatic winds. These results, along with observations

from earlier studies (Sprovieri et al., 2002);(Temme et al., 2003);(Pfaffhuber et al., 2012); (Angot et al., 2016b);(Angot et al., 2016a), demonstrate that, in Antarctica, the inland atmospheric reservoir can influence the cycle of atmospheric Hg at a continental scale. Observations at DDU also highlighted that the Austral Ocean is a net source of GEM in summer and a net sink in spring, likely due to enhanced oxidation by halogens over sea-ice covered areas.

### 4.2.4 Seasonal Patterns Analysis in the Tropical Zone

Relatively few observations of atmospheric Hg had been carried out between the Tropics, before the start of GMOS. Until recently atmospheric Hg data for the tropics were only available from short term measurement campaigns. To date, therefore, there is no information in the Tropical area that can be used to establish long-term trends. Observations in this region may provide a valuable input to our understanding of key exchange processes that take place in the Hg cycle considering that the Inter Tropical Convergence Zone (ITCZ) passes twice each year over this region and the northern and southern hemispheric air





masses may well influence the evolution of Hg concentrations observed in this region. As can be seen in Figure 3, five GMOS sites are located in the Tropics, including Sisal (SIS) in Mexico, Nieuw Nickerie (NIK) in Suriname, Manaus (MAN) in Brazil, Calhau (CAL) in Cape Verde and southern Kodaikanal (KOD) in southern India. GEM concentrations observed in 2013 and 2014 at all sites are comparable with Hg levels recorded at remote sites in the Southern Hemisphere (1.1 to 1.3 $ng\ m^{-3}$ ,

Lindberg et al., 2007). Among these sites, the Kodaikanal site (KOD) shows the highest monthly mean GEM concentrations (see Figures 6 and 7 as well as Table 2 and 3) ranging between 1.25 $ng\ m^{-3}$ (5th percentile) to 1.87 $ng\ m^{-3}$ (95th percentile) during 2013 with an annually-based statistic mean of $1.54 \pm 0.20\ ng\ m^{-3}$ and between 1.20 $ng\ m^{-3}$ (5th percentile) to 2.03 $ng\ m^{-3}$ (95th percentile) during 2014 with an annually average of $1.48 \pm 0.26\ ng\ m^{-3}$ . KOD is a Global Atmospheric Watch (GAW) regional site which is operated by the Indian Meteorological Department. It is worth to point out that the other

tropical GMOS sites are close to sea level and on the coast, whereas KOD is a high altitude site (2333 m a.s.l.). Therefore different meteo-climatic conditions influence the long range transport of air masses to this site. This site is also influenced by anthropogenic sources such as the well-known, but not close, Hg thermometer plant at Kodaikainal (Karunasagar et al., 2006). Due to this anthropogenic influence atmospheric Hg concentrations from 3 $ng\ m^{-3}$ to 8 $ng\ m^{-3}$ for the years 2000 and 2001 have been reported (Rajgopal and Mascarenhas, 2006). India is the third largest hard coal producer in the world

after the People's Republic China and the USA (Pirrone et al., 2010);(Mason, 2009). For the past three decades, India has increased the production of metals, cement, fertilizers and electricity through burning of coal, natural gas and oil becoming one of the most rapidly growing economies (Choi, 2003);(Karunasagar et al., 2006). Relatively little attention has been paid to potential Hg pollution problems due to mining operations, metal smelting, energy and fuel consumption which could impact on ecosystem health (Mohan et al., 2012). Hg concentrations are in fact enhanced in India due to industrial emissions of Hg

mostly from coal combustion (the major source category (48%), followed by waste disposal (31%), the iron and steel industry, chlor-alkali plants, the cement industry, and other minor sources (i.e., clinical thermometers) (Mukherjee et al., 2008);(UNEP, 2008). Unfortunately, details of Hg emissions from these facilities and atmospheric Hg data in general are scarce. Therefore it is necessary for India to generate continuous data, which can be used by scientists for modelling applications to improve emission inventories in order to prevent inaccurate assessments of Hg emission and deposition.

GEM levels observed at Sisal (SIS), Mexico, were below the expected global average concentration ($\sim 1.5 ng\ m^{-3}$). Monthly mean GEM concentrations ranged between 1.0 to 1.47 $ng\ m^{-3}$ in 2013 with an annual average of $1.20 \pm 0.24\ ng\ m^{-3}$ (5th and 95th percentile 0.8 and 1.58 $ng\ m^{-3}$), whereas in 2014 the range varied from 0.82 to 1.45 $ng\ m^{-3}$, with an annual average of $1.11 \pm 0.37\ ng\ m^{-3}$ (5th and 95th percentile 0.82 and 1.45 $ng\ m^{-3}$). GEM measurements at SIS showed in addition, very little variability over the sampling period, indicating that this relatively remote site on the Yucatan Peninsula was not subject to any

significant anthropogenic sources of Hg at all. During 2013 and 2014, the SIS site was typically influenced by the marine air originating from the Atlantic Ocean before entering the Gulf of Mexico. Average GEM concentrations reported at SIS are lower than those recorded in other rural places in Mexico, such as Puerto Angel (on the Pacific coast in Oaxaca state) and Huejutla (a rural area in the state of Hidalgo), where average values of 1.46 and 1.32 $ng\ m^{-3}$ were determined, respectively (De la Rosa et al. 2004). Low GEM concentrations were recorded in 2013 during the later part of the wet season (July/October). Those values

may indicate a slight decrease probably due to deposition processes since the site is a coastal station and subject to frequent

episodes with high humidity caused by rain (Sprovieri, 2016). These findings have also been confirmed through wind roses and backward trajectories that show the predominant wind direction from east-south-east most of the time and sometimes from east-north-east (Atlantic Ocean) (Sprovieri, 2016). In addition, the ITCZ moves north of the equator passing over the Yucatan peninsula during the northern hemisphere summer, causing tropical rain events which could contribute to the slight decrease

of Hg concentrations. Highest GEM levels were observed during the winter period (Dec-Jan) in 2013, whereas 2014 had the lowest GEM concentration in January and higher GEM levels during spring and summer. The background Hg concentrations measured at Sisal are closely comparable to those recorded at Nieuw Nickerie (NIK), Paramaribo, Suriname, located on the north-eastern coast of the South American continent, the first long-term measurement site in the tropics which has been in operation since 2007 (Muller et al., 2012). Analysis of data shows that the annual mean GEM for 2013 and 2014 at NIK are a

little lower than those at SIS, $1.13 \pm 0.42\ ng\ m^{-3}$ and to $1.28 \pm 0.46\ ng\ m^{-3}$, respectively (see Tables 2 and 3). NIK is also a background site because most of the time the air masses arriving at the site come from the clean marine air of the Atlantic Ocean and the influence of possible local anthropogenic sources and continental air is minimal. As the ITCZ crosses Suriname twice each year, the NIK site samples both northern and southern hemispheric air masses. Occasionally higher values are seen, $1.57\ ng\ m^{-3}$ in Feb/Mar 2013 and 1.51 in Aug/Sep 2014. Manaus (MAN) in Amazonia (Brazil) is a GMOS Master site lo-

cated in the Amazon region, an area with a history of important land use change and significant artisanal and small-scale gold mining activities since the 80s. Burning of natural vegetation to produce agriculture lands or pastures represents an important diffuse source of Hg to the atmosphere in Brazil (Lacerda et al., 2004);(do Valle et al., 2005). The analysis of atmospheric Hg species at this site is thus important for the determination of the dynamics of atmospheric Hg. Annual mean Hg concentrations in 2013 and 2014 at MAN are slightly lower than those at both SIS and NIK, with little variability between the two years, see

Table 2 and 3. The measurements from MAN station may therefore suggest that the emissions of Hg from regional biomass burning and ASGM may not have a significant impact locally, but contribute to the global Hg background (concerning Hg from biomass burning see (De Simone et al., 2015)). Unfortunately the emissions from both these sources are associated with large uncertainties and vary over time. Quantifying their impact in South America is extremely important and there is a strong case for expanding the number of GMOS measurements site in the region. Monthly mean GEM concentrations at MAN ranged in

fact, between 1.01 to $1.18\ ng\ m^{-3}$ in 2013 and in 2014 between 0.94 to $1.10\ ng\ m^{-3}$. Also PBM and GOM recorded during 2013 show little variation and varied between 1.35 and $12.70\ pg\ m^{-3}$ (5th and 95th percentile, respectively) with a median value of $3.17\ pg\ m^{-3}$. In 2014, the range was from 0.53 to $5.24\ pg\ m^{-3}$ (5th and 95th percentile, respectively) with a median value of $1.48\ pg\ m^{-3}$. The MAN Hg concentrations therefore seem not to be influenced by regional emissions. However, a number of parameters, such as the intense air mass convection occurring in the Amazon basin and meteorological condition

in general clearly contribute to the observed Hg concentrations, and they do not necessarily reflect only regional emissions (Artaxo et al., 2000);(do Valle et al., 2005).

The Cape Verde Atmospheric Observatory, Calhau Station (CAL) contributes data from the Eastern tropical Atlantic Ocean, where GMOS provides the only existing data set. CAL is an important GAW station located on Sao Vicente Island, approximately 50m from the coastline. GEM measurements from 2012 to 2014 were broadly consistent with previously published

oceanographic campaign measurements in the region, with typical Hg values between 1.1 and $1.4\ ng\ m^{-3}$. The prevailing



wind was from the northeast open ocean bringing air masses from the tropical Atlantic and from the African continent. Due to its relatively long residence time in the atmosphere, the ground level background GEM concentration tends to be relatively constant over the year in tropical regions, unlike mid-latitude and polar regions where a more noticeable seasonal variation has been observed. When compared with measurements from cruise campaigns from North to South Atlantic, we can see that the GEM data at CAL are similar to previously reported southern Atlantic data, where Hg concentrations are lower than the northern part of the Atlantic. Monthly mean GEM concentrations in 2013 ranged between 1.12 to 1.38 $ng\ m^{-3}$ , with an annually-based mean of 1.22 ± 0.14 $ng\ m^{-3}$ (5th and 95th percentile equal to 1.04 $ng\ m^{-3}$ and to 1.46 $ng\ m^{-3}$, respectively), whereas in 2014, the monthly mean observed varied from 1.12 $ng\ m^{-3}$ to 1.33 $ng\ m^{-3}$ with an annually-based mean of 1.20 ± 0.09 $ng\ m^{-3}$ (5th and 95th percentile equal to 1.08 $ng\ m^{-3}$ and to 1.36 $ng\ m^{-3}$, respectively). The highest GEM concentrations in air originating from central Africa have been recorded at CAL when the relative humidity was lowest (occasionally during dust events) (Carpenter, 2011). All Tropical GMOS sites show little atmospheric Hg variability through both the years (2013 and 2014) with small GEM fluctuations during the months which well agrees with a relatively long atmospheric lifetime of Hg in the background troposphere and small variations in the source strength (Ebinghaus et al., 2002) however, clear diurnal cycles of Hg have been conversely observed.

# 5 Conclusions

The higher Hg concentrations and its spatio-temporal variability observed in the Northern Hemisphere compared to the Tropical area and Southern Hemisphere confirms that the majority of emissions and re-emissions are located in the Northern Hemisphere. The inter-hemispherical gradient with higher TGM concentrations in the Northern Hemisphere has remained nearly constant over the years, and confirmed by the observations carried out in the Southern Hemisphere and other locations where before GMOS Hg measurements were lacking or absent. The results of all cruises carried out over the oceans highlighted that in the Northern Hemisphere TGM mean values are almost generally higher than those obtained in the Southern Hemisphere, with a rather homogeneous distribution of GEM in the Southern Hemisphere. The variation of Hg concentration shows increased amplitude in areas strongly influenced by anthropogenic sources. There are, however, uncertainties in the emission estimates especially for the tropical region and the Southern Hemisphere, and not enough long-term information in either areas to identify long-term trends. Long-term atmospheric Hg monitoring and additional ground-based sites within the GMOS global network are important in order to provide datasets which can give new insights and information about the worldwide trends of atmospheric Hg. The over-arching benefit of this coordinated Hg monitoring network would clearly be the production of high-quality measurement datasets on a global scale useful in developing and validating models on different spatial and temporal scales. The GMOS objective of establishing a global Hg monitoring network was achieved always bearing in mind not only the necessity to provide intercomparable data worldwide but also to meet international standards of intercomparibility. In particular, GMOS attempt to comply with the data sharing principles set by the Group on Earth Observations (GEO) aiming to develop the GEOSS encompassing: "observation systems: which include ground-, air-, water- and space-based sensors, field surveys and citizen observatories. GEO works to coordinate the planning, sustainability and operation of these systems, aiming





to maximize their added-value and use; and... information and processing systems: which include hardware and software tools needed for handling, processing and delivering data from the observation systems to provide information, knowledge, services and products." In 2010 the Executive Committee of GEO selected GMOS as a showcase for the Workplan (2012-2015) to demonstrate how GEOSS can support Convention and Policies as well as pioneering activity in environmental monitoring us-

5  ing highly advanced e-infrastructure. Currently GMOS is targeted as the future flagship in the GEO Strategic Plan (2016-2025). The experience gained during GMOS, the development of SOPs for Hg monitoring and the establishment of the Spatial Data Infrastructure (SDI), http://www.gmos.eu/sdi/ (along GEOSS lines), which includes the GMOS Data Quality Management System provide a template to aid countries complying with the requirements of the Article 22 of the Minamata convention.

*Acknowledgements.*  The number of people and institutions to acknowledge for their great contribution during field studies is quite significant

10  for a project like GMOS, however we greatly acknowledge the European Commission for funding GMOS as part of the FP7 (contract no. 26511). Special thanks to those research scientists and technicians that helped to set up aircraft-based platform for UTLS measurements, ground-based sites in remote locations including EV-K2 in the Karakorum, Amsterdam Island, Dome C in Antarctica and all those involved in GMOstral 1028 (IPEV) project for field logistics and technical support.





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
