# Peer review of "Atmospheric Mercury Concentrations observed at ground-based monitoring sites globally distributed in the framework of the GMOS network"

_Atmospheric Chemistry and Physics, 2016_

## Referee Comment (RC1) · J. Pacyna (Referee) · 14 Jun 2016

This is a very important paper presenting the unique ste of data from the GMOS project. The unique character of the data relates to geographical location of more 40 stations of which the data are reported, as well as the various topography of the stations and different climatic zones. The GMOS station network should be seen as the basis for a major monitoring network needed for the assessment of the implementation of the UN Minamata Convention on merucry. The reported set of data forms also a uniquein-formation for validation of regional and global modeling of mercury within air masses.

The major achievements of the GMOS project reported in the reviewed paper include: - establishment of SOG (Standard Operating Procedures) for measuring concentrations of Hg in air samples. This is an important result in the context of the Minamata Convention monitoring network that would need to be established soon, - elaboration of Qa/QC procedures to be further used in the above mentioned monitoring network, - contribution of the GMOS project results to GEOSS, - confirmation of the Northern to Southern Hemispheric gradient of Hg concentrations in the air, indicating an important contribution of anthropogenic emissions to the measures concentrations, - confirmation of seasonal changes of Hg concentrations worldwide. Conclusions presented in the reviewed paper are very important for improvement of our understanding of Hg behavior after its emission to the air. Thes conclusions are fully supported by data presented in the paper. In conclusion, the reviewed paper is a very important contribution to the development of monitoring network needed for the Minamata Convention.

---

## Referee Comment (RC2) · T. Dvonch (Referee) · 17 Jun 2016

This manuscript represents an immense body of effort and successful coordination among research institutions to greatly inform our understanding of the global distribution of atmospheric mercury concentrations. Results from the project importantly confirm a large inter-hemispheric gradient from north to south, while also providing a valuable measurement-based data set which can be used to validate atmospheric mercury models for scenario analysis in support of further mercury research and policy. It is also important to note the degree of detail that went into standardizing SOPs

and data QA/QC procedures, as this assures inter-comparability across the global data network, a large accomplishment considering over 40 research monitoring sites across many research institutions.

Suggested revisions: 1) While very impressive that the GMOS project has been highlighted by GEO as a flagship for future activities, it seems odd that this discussion of GEO is only discussed for the first time in the Conclusions section of the manuscript. Perhaps the detail of this accomplishment with GEO is more appropriate to be discussed earlier in the paper, with a more concise mention then also included as part of Conclusions. 2) The manuscript would benefit from another close proof-read for typos, especially due to the large amount of data description included (for example, page 11, line 12 – seems it instead should read "XN>XT>XS").

---

## Short Comment (SC1) · 23 Jun 2016

Dear Referee, thank you very much for your effort in the reviewed the manuscript on the atmospheric mercury concentrations observed in the framework of the GMOS global network. As you highlighted, the GMOS overarching objective was to establish a global Hg monitoring network having in mind the need to assure high-quality observations in line with international QA/QC standards and to fill the gap in terms of spatial coverage of measurements in the Southern Hemisphere were data were lacking or not existing. One of the major outcomes of GMOS has been an interoperable e-infrastructure

developed following the Group on Earth Observations (GEO) data sharing and inter-operability principles which allows to provide support to UNEP for the implementation of the Minamata Convention (i.e., Art.22 to measure the effectiveness of measures). Thank you very much once more for your important comments on the present paper. Best Regards Francesca Sprovieri

---

## Author Comment (AC1) · 24 Jun 2016

Dear Referee, thank you very much for your comments and effort in the reviewed the manuscript on the atmospheric mercury concentrations observed in the framework of the GMOS global network. Following your suggested revisions, I corrected what you highlighted for typos (pag. 11, line 12) including the revised version of the sentence, i.e., with "XN>XT>XS" (please see pag. 14, line 4). Thank you very much once more. In addition, I agree with you concerning the need to discuss in previous sections of the paper the important results obtained with the GMOS project in supporting the overall

objectives of GEO which in turn highlighted GMOS as a flagship for future activities. Therefore, during the suggested revision step of the manuscript, the accomplishment of GMOS with GEO has been earlier discussed in the manuscript. Please, see pag. 10 line 13 and line 25. Thank you very much once more for your important comments on the present paper. Best Regards Francesca Sprovieri

Please also note the supplement to this comment:

[revised manuscript text omitted]

---

## Referee Comment (RC3) · Anonymous Referee #3 · 12 Jul 2016

Review of the paper ACP2016-466 "Atmospheric Mercury Concentrations observed at ground-based monitoring sites globally distributed in the framework of the GMOS network" by F. Sprovieri et al.

The paper describes the global mercury observation network that has been established in the framework of the EU-FP7 project GMOS in recent years. Observations in the years 2013 and 2014 have been statistically evaluated for the Northern and Southern Hemisphere as well as for the Tropics. The main results are presented and some special features are highlighted.

[Figure]

The paper is worth to be published in ACP, however it needs some significant revisions with regard to the statistical evaluation. My main concern is the data coverage which is very different among the stations and also from year to year. Many conclusions are based on data sets that cover different times or very different number of observations (e.g. 2013/2014 at Dumont d'Urville). Some results look unreliable, e.g. 2013/2014 differences at Mt. Ailao which are not discussed although the Asian stations are intensively described in section 4.2.2.

In addition, there are some contradictions and redundancies that need to be corrected as well as numerous technical errors. These technical errors should have been corrected by the authors before they submitted their manuscript for review.

All pages and line numbers refer to the pdf version of the manuscript.

Major comments:

page 3, line 21: "... and highlight its potential to support the validation ...". I cannot see where this is explained, shown or highlighted in the paper.

Table 1 contains 27 stations, Figure 1 shows 26 stations and Table 2 includes 23 stations. You should explain why this is the case.

Table 2 and 3 and Fig. 1, 2 and 3: It is necessary that you give more information about the data coverage and how the averaging and statistical evaluation has been performed. For the monthly average: how many days were used to generate this average or is this the average of all observations within the month? For the annual average: Is this the average of all observations (each lasting 300 s) or did you first calculate daily averages and then the annual average? You should always give the number of data points that is behind the values you have in the tables and in Fig. 3. One of the main shortcomings of this paper is that at some stations data coverage is very inhomogeneous (at least that's what I get from Fig. 1) and therefore annual averages might not be comparable between the years 2013 and 2014.

[Figure]

Table 3: The concentrations at MAL are much higher in 2013 compared to 2014. Is there a good reason for this?

page 9, section 4.1: Here, you should give more information about data coverage and consistency, e.g. about the averaging methods, the number of available data points etc. In line 21 you mention that most measurements started at the end of 2011 and your evaluation is for 2013 and 2014. What about 2012?

page 9, line 25: Tables 2 and Tables 3 do not contain all GMOS sites as I mentioned earlier. The stations in Fig. 1 are not consistent with those in Table 1 and Table 2.

page 9, line 26/27: You need to be much clearer with averages, means and medians. What you call "mean concentrations" are station averaged medians. How were the annual medians derived? What is given in Table 3 and what is the basis for the values?

page 11, line 5-8: You say that you fitted a log-normal distribution. Why did you do so and how did you do it? The PDFs in Fig. 4 and Fig. 5 look very much like normal distributions. Can you show that the frequency distribution of the observations has a skewness that differs from zero? The standard t-Test is only applicable to normal distributions, which might be fine if I look at the PDFs. However, you claim that the concentration values are log-normally distributed. In my opinion you test if the means are different on a 99% confidence level. Again, you need to say what the basis for your evaluation and your fit is. Did you simply take all 300s observations from all stations in the individual sub-groups (Northern, Tropics, Southern)? Why do you show the same for the monthly averages?

page 13, Table 5 and page 17, lines 8-11: I think it would make more sense to give a constant p-value (say 0.05, the most commonly used value) and then give the confidence intervals. This could result in the means of MCH and MAL being not significantly different on this level. I wonder why the difference between MCH and MWA should be lower than that between MCH and MAL. This looks wrong in the table.
I also cannot follow the explanation on page 17 (l 8-11), that tries to demonstrate that

the PDFs of MCH and MWA are significantly different. I do not see why this is to be shown. It is certainly not true on the p = 0.05 level.

page 15, lines 31-34: It would be nice if you could elaborate a bit more about what would be needed in order to understand the fate of atmospheric Hg and the reaction kinetics. Certainly global GEM observations are valuable but not enough. This could of course also be done in the conclusions.

page 22, lines 20-22: Shouldn't it be visible in the observations at MAN if there is an influence from regional sources on the mean concentrations? Did you analyse temporally higher resolved data than monthly averages?

page 22, lines 28-31: These statements are very weak. You do not explain how the meteorological conditions influence the observed concentrations. Obviously, GEM concentrations are always influenced by the hemispheric background. The questions is why you do not see a regional impact from the sources that are expected to be present in the area. Can that be answered by the meteorological conditions?

page 23, line 29 - page 24 line 8: These statements about GEO and GEOSS do not fit into the conclusions. GEO is briefly mentioned before on page 4. Some of this text about GEO would fit better there.

page 23/24, Conclusions: To me, the conclusions sound too general. You should say more about what you found out with the analysis presented in the paper and then give an outlook about what can be achieved if the observations are continued or improved.

Minor comments:

page 4, line 7: Where are the guidelines available? Is there a web page where they can be read or downloaded?

Table 1: Put the explanations in the bottom into the caption. Give the "Country" as third column right to "Site". Give units for Lat and Lon. Replace ',' with '.' (e.g. -37.79604).

page 6, line 34: what exactly are sub ng/m$^{-3}$ levels. This can be much if the concentrations are not higher than 1 ng/m$^{-3}$.

page 7, line 1: How is it possible that GMOS results have been published in 2010 when the project started in November 2010 as you indicate here? Is there another reference to GMOS intercomparison studies?

Table 2: The percentiles in this Table are also shown in Fig. 3. They could be moved into an appendix. The caption would need more explanations, e.g. what "5th", "25th", ... means. Mean and st. dev might then be moved to Table 3.

Table 3: You need to give more explanations in the caption about what is shown. "Monthly based statistics" does not tell much. What is shown? What are the units? Are these mean or median values?

page 10, Fig. 1 and Fig. 2: The details are hard to read because the pictures are too small. Why do you say "some of the ... stations". This figures contains more stations than Table 1. Some of them, e.g. Iskrba are not used anymore. Why?

page 10, line1: "according to their location": What is the rule for this? I suppose by latitude from North to South but it is not mentioned.

page 11, Fig. 3: This figure contains the same information as Table 2.

page 11, line 4: I suppose the groups are related to latitude (not longitude).

page 12, Fig. 4: This figure could be smaller. How did you choose the width of the bins for the histograms that represent the observations?

page 12, Table 4: Sometimes you give 3 and sometimes 4 significant digits. What is the reason behind this?

page 13, caption of Table 5: What are "experimental measures"?

page 15, line 6-8: You say that concentrations at EVK are comparable to aircraft observations in August 2013 over Europe. The aircraft observations represent just a snapshot. Do you want to argue that GEM concentrations in the free troposphere are always around 1.3 ng/m$^3$? If this is the case you need to present more evidencde for this.

page 17, line 31: Is there any proof for higher direct PBM emissions in winter? What are the sources

page 17, line 34 - page 18, line 1: Is there a proven relation between high PM2.5 concentrations and high PBM concentrations as you indicate there?

page 18, line 2: How do you know that high GOM values were due to local sources. Is there anything that supports this?

page 19, lines 1/2: How do the observations support the fact that East Asia is the biggest source region the world?

page 19, line 5: Again, I get confused with the median and mean GEM values. You should only use one of those, depending on the frequency distribution of the observations.

page 19, line 19/20: "... anthropogenic emissions from ... can sometimes be observed": Is this shown somewhere or is there a reference for this?

page 19, lines 31/32: "... the lower concentrations ... were associated with ...": Again, you claim something that cannot be verified by the reader through an analysis that you performed. You don't give any reference, either.

page 20, line 7: "..., reported concentrations ...": Please give the reference where they were reported.

page 20, line 23/24: "... exported from ... by strong katabatic winds.": Did you investigate this? How is it proven? Or is it taken from another publication? Then it should be cited, here.

page 20, line 23: "Observations at DDU also highlighted ... ": The reader cannot follow this. Where can this be seen?

page 21, lines 1/2: What about Celestun? According to Table 1, this is in the tropics, too.

page 21, line 11: What do you mean with "meteo-climatic conditions"?

page 21. line 12. How close is the thermometer plant?

page 21, line 14: " India is the third largest hard coal producer ...": Concerning Hg emissions, it is more important how much is consumed.

page 21, line 23: "... it is necessary for India ...": I think this necessary at other places in the world, too.

page 21, line 30/31:"SIS site was typically influenced by ...": Again you claim something that the reader is not able to follow. This leaves the impression that you speculate or that it is published somewhere else and you do not give a reference.

page 22, line 11/12: Please explain how you analysed the influence of different air masses on the concentrations measured at NIK.

page 22, line 34: Here you refer to the year 2012 while all other stations were only evaluated for 2013/2014.

page 22, line 35, page 23, line 1: Again, the reader gets the impression that you analysed the meteorological conditions at the different stations but you do not explain how you did this.

page 23, line 18/19: Over which years has the inter hemispherical gradient been constant? You analyse only two year in this paper. Or do you refer to other investigations?

page 23, line 20: You mention cruises but there is no data from cruises presented here.

Technical corrections:

affiliation 14: Sweden

Write all units correctly, e. g. on page 2, line 24: convert ng m-3 into $ng\,m^{-3}$. This appears at several places in the entire document.

Improve the way citations are shown, by removing inner brackets when two or more references are given, e.g. page 2, line 23: (Lindberg et al., 2007; Sprovieri et al., 2010b). This appears at several places in the entire document.

Explain TGM earlier in the document. It appears first on page 3, line 32 but is explained later.

Take care about words starting with capitals. E.g. on page 4, line 2, I propose to write "GMOS external partners". There are several other places where the use of capitals should be re-considered.

Explain PBM2.5 on page 4.

Give more explanations about what is displayed in Table 1 and Table 2 in the respective captions.

The graphs in Fig. 1 and Fig. 2 are too small and therefore hard to read.

page 11, line 4: "latitude" instead of "longitude".

Fig. 4 and Fig. 5: If one has a title the other should also have one. They could be a bit smaller or combined into one figure with two panels. It is not clear what "raw data" in the caption of Figure 4 really means.

Table 5 looks misplaced, it is first mentioned 5 pages later. The number 0.0.365 is misspelled.

Figure and table captions should end with a full stop.

Abstract, page 1, line2: understand

page 2, line 30: Wängberg

page 3, line 14: and a QA/QC ...

page 4, line 15: ... at the French site Dumont d'Urville.

page 4, line 29: ... within the GMOS network.

page 6, line 4: This is also in line with a study recently published by Slemr et al. (2015) ...

page 6, line 8: located at the coastline

page 9, line 20: raw data

page 13, Table 5: 0.365. The table is misplaced because it is mentioned much later in the text (page 16).

page 15, line 12/13: misplaced brackets for the references.

page 15, line 15: Which station is STN?

page 17, line 3/4: ... the three distributions come from the same ...

page 17, line 7: Table 5 is on page 13, which is not close enough to the page where it's mentioned.

page 17, line 23: ... during the years 2013 ...

page 17, line 24: I think this has to be GOM (not GEM).

page 18, line 7: ... at the edge of the north-eastern part ...

page 18, line 9: Hg sources

page 20, line 17: in the same period

page 20, line 30: in the Tropics

---

## Author Response (AR1)

**Reply to Comments from: T. Dvonch (Referee #2)**

Dear Referee,

First of all, thank you very much for your comments and effort in the reviewed the manuscript on the atmospheric mercury concentrations observed in the framework of the GMOS global network.

Following your suggested revisions:

1) While very impressive that the GMOS project has been highlighted by GEO as a flagship for future activities, it seems odd that this discussion of GEO is only discussed for the first time in the Conclusions section of the manuscript. Perhaps the detail of this accomplishment with GEO is more appropriate to be discussed earlier in the paper, with a more concise mention then also included as part of Conclusions.

**Reply**: We agree with you concerning the need to discuss in previous sections of the paper the important results obtained with the GMOS project in supporting the overall objectives of GEO which in turn highlighted GMOS as a flagship for future activities. Therefore, during the suggested revision step of the manuscript, the accomplishment of GMOS with GEO has been earlier discussed in the manuscript. This issue has also been highlighted by the 3$^{rd}$ Referee, therefore, Please, see the revised manuscript at page 4 line 16-25.

2) The manuscript would benefit from another close proof-read for typos, especially due to the large amount of data description included (for example, page 11, line 12 – seems it instead should read "XN>XT>XS").

> **Reply**: Yes, thank you. We corrected what you highlighted for typos at page 11, line 5 with "XN>XT>XS". Thank you very much.

Thank you very much once more for your important comments on the present paper.

**Reply to Comments from: Anonymous (Referee #3)**

Dear Referee,

First of all, thank you very much for your effort and useful comments reported within the reviewed manuscript on the atmospheric mercury concentrations observed in the framework of the GMOS global network. We followed through the manuscript your major, minor and the technical comments made, reviewing the manuscript according to taking into account your suggestions. We think that after the detailed review our manuscript has been now improved. Please, see below our reply to your comments, section by section from the major comments to the technical corrections. Thank you very much once more.

**Major Comments:**

1. page 3, line 21: "... and highlight its potential to support the validation ...". I cannot see where this is explained, shown or highlighted in the paper.

**Reply:** In the sentence line 21, page 3 (now page 3, line 31) we would emphasize the importance of a global network to provide high-quality measurement datasets which can give new insights and information about the worldwide trends of atmospheric Hg with significant implications for refining existing regional and global models, and developing new ones as well as model/measurement intercomparison, validation and so on. In this perspective, the datasets represent a potential support to modeling studies on mercury process and deposition on environmental ecosystem. This is explained now within the revised conclusions following also your suggestion n. 10. Please, see the revised section "Conclusions". Thank you once more for your input.

2. Table 1 contains 27 stations, Figure 1 shows 26 stations and Table 2 includes 23 stations. You should explain why this is the case.

**Reply:** Table 1 and Figure 1 show the core stations that are part of the GMOS network that sent on regular basis the raw data (data on which the QA/QC process has not been yet applied) to the central GMOS database, and currently are sending the data in near real time mode. I'm sorry, Figure 1 missed the French site "La Seyne-sur Mer"(LSM) which now has been included within the Fig.1, therefore, Table 1 and Figure 1 show both the same number of stations. Please, see Figure 1 at page 9. Thank you for your input. Regarding Table 2 (now Table SM1 reported within the Supplementary Material), it shows a different number of stations (n. 23) because our discussion is mostly related to the 2013 and 2014 years which as specified earlier in the manuscript, is the period with a higher % of data coverage. Table SM1, in fact, shows statistical information for 2013 and 2014 yrs on the QA/QC data validated within the central system, therefore, during the validation process, we considered only the high quality data coming from the stations which result in a number of 23 monitoring sites. The other stations whose high quality data are scarce for the period considered have been ruled out from the calculation and discussion of results. If you need more information about the validation process of the GMOS central system, please, see the manuscript, previously published: D'Amore et al., 2015, (*D'Amore, F., Bencardino, M., Cinnirella, S., Sprovieri, F., and Pirrone, N.: Data quality through a web-based QA/QC system: implementation for atmospheric mercury data from the Global Mercury Observation System, Environmental Sciences: Processes and Impacts, 17, 1482–1492, doi:10.1039/C5EM00205B, 2015.*). Thank you.

3. Table 2 and 3 and Fig. 1, 2 and 3: It is necessary that you give more information about the data coverage and how the averaging and statistical evaluation has been performed. For the monthly average: how many days were used to generate this average or is this the average of all observations within the month? For the annual average: Is this the average of all observations (each lasting 300 s) or did you first calculate daily averages and then the annual average? You should always give the number of data points that is behind the values you have in the tables and in Fig. 3. One of the main shortcomings of this paper is that at some stations data coverage is very inhomogeneous (at least that's what I get from Fig. 1) and therefore annual averages might not be comparable between the years 2013 and 2014.

**Reply:** Thank you for your comment. As Table 2 is big in size we decided to move it in a "Supplementary Material Section"as Table SM1. We integrated Table 2 (now Table SM1) giving more information on the number of data points that is behind the values we reported in this Table, on annual basis and for 2013 and 2014, respectively. Both monthly and annual averages were generated taking into account all observations within the month and the year, respectively. We generated an additional Table (see Table SM2) also reported in the Supplemental Section where we reported only the number of data points on monthly basis and for 2013 and 2014, respectively.In generating both monthly and annual statistics we considered the maximum time resolution available at each sampling site and, for clarification, we reported on both monthly and annual basis, an additional column reporting the maximum time resolution available. Please, seethe new Tables reported within the Supplementary Material.

4. Table 3: The concentrations at MAL are much higher in 2013 compared to 2014. Is there a good reason for this?

**Reply:** Yes, valid comment related to Table 3 (now Table SM2 in "Supplementary Material"). Thank you. We discussed the possible reasons of atmospheric Hg decrease from 2013 to 2014 yrs at Mt. Ailao with the site managers as well as with professor Feng and professor Fu, and they inferred that the decrease of coal consumption in China and forest fires in southwestern China and southeastern Asia might be the possible reason of atmospheric Hg reduction from 2013 to 2014 (personal communication). As you know in China there are several illegal activities (i.e., gold mining) which might influence Hg concentrations, however, in this case, there are no available documented reports and/or papers which can surely explain this decrease observed. This is also discussed in Pirrone et al., 2016 (paper in preparation for ACP-Special Issue). Thank you for your comment.

5. page 9, section 4.1: Here, you should give more information about data coverage and consistency, e.g. about the averaging methods, the number of available data points etc. In line 21 you mention that most measurements started at the end of 2011 and your evaluation is for 2013 and 2014. What about 2012?

**Reply:** Thanks for pointing out this issue. We reported more information about data coverage and consistency according to your comment in the revised section 4.1.Please see the revised section 4.1 at page 8 from line 20.

In line 21 (now line 24, section 4.1) we revisedthis part of the section because in 2011 (at the effectively starting of the project) only four monitoring sites produced Hg measurements, and step by step, an increasing number of stations have been established and added to the network in 2012. Therefore, we evaluated the two years (2013/2014) due to major data coverage (%) of the observations. In fact, our statistical evaluations/calculations are related to this period for all the ground-based sites taken into account within the GMOS network in order to harmonize the

discussion and compare the results worldwide. However, this not exclude the possibility that somewhere in the manuscript we reported some comments on observations obtained in 2012 at some stations if they add value to the discussion of the results. Please, see the revised Section 4.1, line 24 – 30.Thank you once more for your comment.

6. page 9, line 25: Tables 2 and Tables 3 do not contain all GMOS sites as I mentioned earlier. The stations in Fig. 1 are not consistent with those in Table 1 and Table 2.

**Reply:** Thank you for taking care this important point. Please, see our reply to your 2nd comment.

7. page 9, line 26/27: You need to be much clearer with averages, means and medians. What you call "mean concentrations" are station averaged medians. How were the annual medians derived? What is given in Table 3 and what is the basis for the values?

**Reply:** Yes, your comment is right. I'm sorry we get confused about the correct statistic words. In each case mentioned at this page, now is page 10, we would mean the average concentrations obtained, for the Northern, Tropics and Southern Hemisphere as the average of the mean values recorded at the stations located within them. Therefore, at page 10, we corrected what you highlighted in your comment. Please, see page 10, line 3-5. Thank you very much.

8. page 11, line 5-8: You say that you fitted a log-normal distribution. Why did you do so and how did you do it? The PDFs in Fig. 4 and Fig. 5 look very much like normal distributions. Can you show that the frequency distribution of the observations has a skewness that differs from zero? The standard t-Test is only applicable to normal distributions, which might be fine if I look at the PDFs. However, you claim that the concentration values are log-normally distributed. In my opinion you test if the means are different on a 99% confidence level. Again, you need to say what the basis for your evaluation and your fit is. Did you simply take all 300s observations from all stations in the individual sub-groups (Northern, Tropics, Southern)? Why do you show the same for the monthly averages?

**Reply:** The comment is right, the PDFs are almost normal. The monthly distribution figure (Fig. 5) was added just to be coherent with all other plots in the manuscript, since every plot is related to monthly averaged quantities. Actually it can be removed as suggested by the referee since it's simply redundant. Concerning the PDFs, obviously the fit is performed on a normalized histogram with unit area of all samples divided in three subgroups. However it's should be noted that the analysis deal with strictly positive values so that a log-normal distribution, with shape factor ~ 0, should be well suited for fitting the experimental values.In addition it's should be pointed out that for large samples size, slightly non-normal distributed the T-Test gives also a robust estimates (for major clarification, please, see for example, the reference: *J. L. Devore and K. N. BerkModenr mathematical statistics with applications, Springer 2012*). Inthe Figure 4 we plot the difference of the two distributions. However, as requested by the referee, we fitted the experimental data also with a normal distribution. Please, see the new Figure 4 in the manuscript at page 11. The corresponding text of the manuscript has been also revised. Please, see at page 11, line 5 –16, and page 12, line 1- 9. Thank you.

9. page 13, Table 5 and page 17, lines 8-11: I think it would make more sense to give a constant p-value (say 0.05, the most commonly used value) and then give the confidence intervals. This could result in the means of MCH and MAL being not significantly different on this level. I wonder why the difference between MCH and MWA should be lower than that between MCH and MAL. This looks wrong in the table. I also cannot follow the explanation on page 17 (l 8-11), that tries to demonstrate that the PDFs of MCH and MWA are significantly different. I do not see why this is to be shown. It is certainly not true on the p = 0.05 level.

**Reply:** For this second case (strongly non-normal) the core of the PDFs is normal, however the tails must be taken into account. For this reason we perform an alternative test: let us consider a pair of our three time series, namely Xi (i = 1; 2) which corresponds to independent random samples described by the log-normal distributions. Then the random variables Yi = ln (Xi) are close to normal distribution with means $\mu i$ and variances $\sigma^2_i$ , namely Yi ~ N($\mu i$, $\sigma^2_i$). Since $\eta i = \exp(\mu i + 0.5 \sigma 2i)$ is the expectation value for Xi, the problem of our interest is then to test the null hypothesis about $\eta_2 - \eta_1$. More formally, we test H$_0$: $\varnothing \leq 0$ where $\varnothing = \eta_2 - \eta_1$. In other words we test the null hypothesis that there is a significant difference in the sample means. Using the algorithm described in: *K. Krishnamoorthy and T. Mathew, Journal of statistical planning and inference 115 (2003) 103-121*; *K. Abdollahnezhad, M. Babanezhad and A. A. Jafari, Journal of statistical and econometrics methods vol.1 no.2 (2012) 125-131*, specifically designed to perform the inference on difference of means of two log-normal distributions. We obtain the estimates for the p-values which are close to 1 and the confidence intervals, calculated at a confidence level of 95%, which are reported in the new Table 5 (now Table 3)which replaced the old Table 5.Please, see in the manuscript at page17 the new Table 3,and the revised text  from page 15, line 18 to page 16, lines 1 - 29. Thank you.

10. page 15, lines 31-34: It would be nice if you could elaborate a bit more about what would be needed in order to understand the fate of atmospheric Hg and the reaction kinetics. Certainly global GEM observations are valuable but not enough. This could of course also be done in the conclusions.

**Reply:** Done, Thank you. Please, see the new revised conclusions of the manuscript at page 23 and page 24.

11. page 22, lines 20-22: Shouldn't it be visible in the observations at MAN if there is an influence from regional sources on the mean concentrations? Did you analyse temporally higher resolved data than monthly averages?

**Reply:**yes, we analyzed temporally higher resolved data than monthly averages as reported previously. Hg concentrations observed at MAN over 2013-2014 period as reported within the manuscript are mostly uniform with very little variations. Artaxo et al. in a previous study on atmospheric Hg concentrations sampled by aircraft measurements over different sites in the Amazon Basin (and among them also MAN) found Hg concentrations between 0.5 to 2 ngm$^{-3}$ at pristine sites not impacted by air-masses enriched with emissions from gold mining areas and/or biomass-burning which are the most important emission sources of Hg, respectively. Those data collected from August to September, 1995 are comparable to ours observed in 2013 and 2014 at MAN, whereas at other sites over areas with intense biomass burning and near areas with strong Hg emissions (Alta Floresta and Rondonia, for example) they found high Hg levels. The three main gold mining areas in the Amazon basin are located in Rondonia, Mato Grosso and in the South of the Parà states, which are areas all at south of Manaus. In addition, the general air-masses circulation pattern during the sampling period was from the Parà state (east-north-east), following the Amazon, Rondonia, Mato Grosso states and the plume leaves South America in the Southern part of Brazil (Trosnikov and Nobre, 1998; Artaxo et al., 2000). During the GMOS period the prevailing winds during the wet seasons (from Jan-March) were from North-North-East, North-East, and East-North-East, whereas during the dry seasons (from Aug-Oct) were from North and North-North-East as well as North-North-West. It is important to point out, therefore, that the position of the monitoring site (MAN) is in a pristine area of the Amazon Basin located upwind from gold mining areas that as highlighted by Artaxo et al. (2000) represent in Amazon basin the most important emission source of Hg followed by the biomass-burning. However, in order to make clear and improved  this part of the manuscript, we rewrite some sentences adding other information on this as follows:

….. "*The measurements from MAN station may therefore suggest that, although the Hg emissions from regional biomass burning and ASGM represent the major emission sources in the Amazon basin as reported in a study performed by Artaxo et al. (2000), they may not have a significant impact locally, but contribute to the global Hg background (concerning Hg from biomass burning see (De Simone et al., 2015). MAN is in fact, a very remote site, inside the campus of the Embrapa Amazonia oriental and upwind from the three main gold mining areas in the Amazon basin which are located in Rondonia, Mato Grosso and in the South of the Parà states (Artaxo et al., 2000). Previous Hg measurements performed by Artaxo et al. (2000) during an aircraft experiment over different sites in the Amazon Basin highlighted Hg concentrations between 0.5 to 2 ngm$^{-3}$ at pristine sites (and among them also MAN) not impacted by air-masses enriched with emissions from gold mining areas and/or biomass-burning. Those data collected from August to September, 1995 are comparable to ours observed in 2013 and 2014 at MAN during the same period, whereas at other sites over areas with intense biomass burning and near areas with strong Hg emissions (Alta Floresta and Rondonia, for example) reported very high Hg levels (5 – 14 ngm$^{-3}$)(Artaxo et al., 2000). These high Hg concentrations have never observed at MAN during the 2013 and 2014 period…..*
*….Most of the air masses that reach the site in 2013 and 2014 comes from Tropical Atlantic, and travels for about 1,500 Km over pristine forest before reaching the site (Artaxo et al., 2015), and the prevailing winds during the wet seasons (from Jan-March) were from North-North-East, North-East, and East-North-East, whereas during the dry seasons (from Aug-Oct) were from North and North-North-East as well as North-North-West (Artaxo et al., 2015)."…*

Please, see page 21, line 25 - 35 and page 22, line 1-4 and line 11 - 15**).** Thank you for your comment.

12. page 22, lines 28-31: These statements are very weak. You do not explain how the meteorological conditions influence the observed concentrations. Obviously, GEM concentrations are always influenced by the hemispheric background. The questions is why you do not see a regional impact from the sources that are expected to be present in the area. Can that be answered by the meteorological conditions?

**Reply:** Please, see our reply above (to your comment n. 11) and the related changes in the manuscript. Thank you.

13. page 23, line 29 - page 24 line 8: These statements about GEO and GEOSS do not fit into the conclusions. GEO is briefly mentioned before on page 4. Some of this text about GEO would fit better there.

**Reply:** Thank you for your suggestion. We have modified the Conclusions and reported some of the text on GEO/GEOSS you suggested at page 4 according to. Please, see the section "Conclusion" at page 24, line 3 – 13. These sentences were deleted here and replaced at page 4, line 16 - 25.

14. page 23/24, Conclusions: To me, the conclusions sound too general. You should say more about what you found out with the analysis presented in the paper and then give an outlook about what can be achieved if the observations are continued or improved.

**Reply:**The conclusions have been reorganized and modified according to your suggestion and comment reported also earlier. Please, see the conclusion revised at page 23/24. Thank you.

**Minor comments:**

1. page 4, line 7: Where are the guidelines available? Is there a web page where they can be read or downloaded?

**Reply:** Yes, the "Governance and Data Policy of the Global Mercury Observation System" guidelines is a document which is available and downloaded from the GMOS web page: www.gmos.eu and in particular at this following link: http://www.gmos.eu/public/GMOS-Governance_Data_Policy_rev160705.pdf you can read and/or download it. Following your comment, anyway, we have insert within the sentence at page 4, now, line 28 in the manuscript, the reference "Pirrone, 2012" where you can also see the link within the reference section. Thank you.

2. Table 1: Put the explanations in the bottom into the caption. Give the "Country" as third column right to "Site". Give units for Lat and Lon. Replace ',' with '.' (e.g. -37.79604).

**Reply:** Yes, done. Thank you. Please, see revised Table 1 according to at page 5 of the manuscript and the related caption integrated with information needs.

3. page 6, line 34: what exactly are sub ng/m$^{-3}$ levels. This can be much if the concentrations are not higher than 1 ng/m-3.

**Reply:** Thank you for your suggestion. Your comment is related to the Section 3.2, now at page 7. We rewrote the sentence as follow: **…"**The alternative automated instrument to measure continuous GEM concentrations is the Lumex RA-915AM which is based on the use of differential atomic absorption spectrometry with direct Zeeman effect providing a detection limit lower than 1 ng m$^{-3}$ (Sholupov and Ganeyev, 1995; Sholupovet al., 2004)."… Please, see pag. 7, line 21 - 24.

4. page 7, line 1: How is it possible that GMOS results have been published in 2010 when the project started in November 2010 as you indicate here? Is there another reference to GMOS intercomparison studies?

**Reply:**In that sentence, we referred to an intercomparison study performed in the framework of the CEN TC/264 working group on the development of the European Standard Methods on Total Gaseous Mercury in ambient air (EN15852) and in precipitation (EN5853). We reported at the end of the sentence the reference of the work published on this intercomparison work and results. Thank you for your comment because it advise us that the sentence as it has been written could give a reader to misunderstand exactly what it means. Therefore, we rewrote the sentence as:

**…"**Comparison studies between the Tekran 2537 and the RA-915AM performed both during EN 15852 standard development showed good agreement of the monitoring data obtained with these systems (Brown et al., 2010b)."…...

Please, see at page 7, now, line 24 - 26. Thank you.

5. Table 2: The percentiles in this Table are also shown in Fig. 3. They could be moved into an appendix. The caption would need more explanations, e.g. what "5th", "25th", ... means. Mean and st. dev might then be moved to Table 3.

**Reply:** Yes, we agree with you. We moved the percentiles, as well as the mean and St. Dev. generated on annual basis into the Supplementary Material annexed to the revised manuscript. As you suggest we also put some more explanation regarding percentiles in the caption of this Table. Please, see the Table SM2 in the Supplementary Material.

6.  Table 3: You need to give more explanations in the caption about what is shown. "Monthly based statistics" does not tell much. What is shown? What are the units? Are these mean or median values?

**Reply:** Yes, done. Thank you. Please, see the Table SM1 corrected according to and also reported within the Supplementary Material.

7.  page 10, Fig. 1 and Fig. 2: The details are hard to read because the pictures are too small. Why do you say "some of the ... stations". This figures contains more stations than Table 1. Some of them, e.g. Iskrba are not used anymore. Why?

**Reply:** Thank you for your comment and suggestion. We enlarged both Figure 1 and Figure 2 according to. Figure 1 now contains the same number of stations of Table 1 (please, see our reply to your 2$^{nd}$ comment in the section "Major Revision"). Figure 2 refers only to the "Master" stations of the GMOS network that consist to date in a restricted number compared to the Secondary stations. Some few stations reported within Figures and Tables, such as Iskrba have been ruled out the discussion because the data of high quality are not consistent with a serious discussion due to several technical problems they have had during the period chosen for the discussion of the results.

8.  page 10, line1: "according to their location": What is the rule for this? I suppose by latitude from North to South but it is not mentioned.

**Reply:** Yes, that's right, according to their latitude. Anyway, we make more clear the sentence as following:

…"The sites have been organized in the graphic as well as in the Tables according to their latitude from those in the Northern Hemisphere to those in the Tropics and in the Southern Hemisphere"…

Please, see page10, line 8 - 9.Thank you.

9.  page 11, Fig. 3: This figure contains the same information as Table 2.

**Reply**: Yes, we agree, however, we thought that the same information reported graphically can show more clearly the pattern of the results obtained. Therefore, according to your clarification, we thought to replace the Table 2 in a supplement file as "Supplementary Material" of the manuscript leaving into the manuscript only the Figure 3. In the "Supplementary Material" the Table 2 has been target as Table SM1. Please, see the "Supplementary Material" file annexed to the revised manuscript. Thank you.

10. page 11, line 4: I suppose the groups are related to latitude (not longitude).

**Reply:** Yes, corrected. Please, see page 11, line 9 – 10. Thank you.

11. page 12, Fig. 4: This figure could be smaller. How did you choose the width of the bins for the histograms that represent the observations?

**Reply:** yes, we replaced and reduced the new Figure 4 according to. Please, see at page 11 the new Figure 4. Regarding the choose of the bins width we followed the Scott rule.

12. page 12, Table 4: Sometimes you give 3 and sometimes 4 significant digits. What is the reason behind this?

**Reply:** No reason behind. Thanks a lot for your comment. There was a typo, therefore we correct what you mentioned harmonizing the number at three digits. Please, see the new Table 4, now reported as Table 2 at page 12 of the manuscript revised. Thank you once more.

13. page 13, caption of Table 5: What are "experimental measures"?

**Reply:** The "experimental measures" from our point of view are the Hg field data. Anyway, following your comments(N. 9 in "Major Comments" section) we replaced both Caption and the old Table 5 with the "new Table 3" and related new Caption". Please, see the revision done in the manuscript at page 17. Thank you.

14. page 15, line 6-8: You say that concentrations at EVK are comparable to aircraft observations in August 2013 over Europe. The aircraft observations represent just a snapshot. Do you want to argue that GEM concentrations in the free troposphere are always around 1.3 ng/m$^3$? If this is the case you need to present more evidence for this.

**Reply:** Thank you for your comment. Regarding the concentrations observed at EVK, we just highlight that both the mean and median value at EVK are comparable to aircraft observations performed in August 2013 which is a limited period as you rightly say in your comment, therefore, we don't absolutely argue that GEM concentrations in the free troposphere are always around 1.3 ngm$^{-3}$ but just report a comparison with the results observed also during aircraft measurements performed.

15. page 17, line 31: Is there any proof for higher direct PBM emissions in winter? What are the sources

**Reply:** There are several works published by Wang et al., 2006; Fu et al., 2008b; 2012a; 2015; Zhu et al., 2014 which discussed about TPM/PBM concentrations observed in several provinces and areas of China, and among them also the regions of our interest in regarding the Chinese sites that are part of GMOS. They discussed about the higher TPM/PBM observed in winter that in their opinion could be likely caused by direct PBM emissions, formation of secondary particulate mercury via gas-particle partitioning and a lack of wet scavenging processes. Fu et al., 2015, in addition, in this review paper found a positive correlation between GEM and PBM and argued that PBM and GEM shared common emission sources. At Mt. Walinguan, in addition, Fu et al. (2015) also observed elevated PBM concentrations duringnighttime probably caused by downward intrusion of PBM-enriched air originating from regional industrialized and urbanized areas (Fu et al., 2012a).Regarding the sources, they attributed these higher concentrations to coal burning in industry and domestic heating as well as tosmelting of non-ferrous metals (e.g., Zn) which is one of several other important Hg emission sources (see: Feng et al., 2004). The references are reported within the manuscript. Thank you for the comment.

16. page 17, line 34 - page 18, line 1: Is there a proven relation between high PM2.5 concentrationsand high PBM concentrations as you indicate there?

**Reply:** This was discussed within the already published paper in ACP. Please, see the review paper: Fu et al., 2015.

17. page 18, line 2: How do you know that high GOM values were due to local sources. Is there anything that supports this?

**Reply:** GOM due to its chemical-physical characteristics has a much shorter atmospheric residence time and limited long-range transport (Lindberg and Stratton, 1998). However we also taken into account that some meteorological factors, such as low air humidity and high wind speed, the possibility of regional transport of GOM cannot be ruled out.

18. page 19, lines 1/2: How do the observations support the fact that East Asia is the biggest source region the world?

**Reply:** There are several research works done on anthropogenic emission from Asia and East Asia which support this statement. Please, see the reference reported above, within our reply to your comment n. 15 (minor revisions) as well as within the manuscript in the "Reference" section. Please, see also: van Donkelaar et al., 2010, and *Y. Qin and S. D. Xie, Spatial and temporal variation of anthropogenic black carbonemissions in China for the period 1980–2009. Atmos. Chem. Phys., 12, 4825–4841, 2012.*Thank you.

19. page 19, line 5: Again, I get confused with the median and mean GEM values. You should only use one of those, depending on the frequency distribution of the observations.

**Reply:** Yes, we agree with you. We corrected the sentence referring only to mean GEM values. Please, see now at page 18, line 5.Thank you.

20. page 19, line 19/20: "... anthropogenic emissions from ... can sometimes be observed":Is this shown somewhere or is there a reference for this?

**Reply:** yes, there are some previous works performed by Brunke et al., 2004; 2012;Slemr et al., 2015;which discussed about this. These references have been also reported within these sentences in the manuscript. Please, see page18, line 20.Thank you.

21. page 19, lines 31/32: "... the lower concentrations ... were associated with ...": Again,you claim something that cannot be verified by the reader through an analysis that youperformed. You don't give any reference, either.

**Reply:** I'm sorry if we didn't repeat several time the reference "Angot et al., 2014" within this part of the manuscript, probably resulting in more confusion for the referee. These statements represent an analysis performed on results obtained in a research work by Angot et al., 2014 which is inserted atpage19, line 26 and 28 of the manuscript.Thank you for your comment.

22. page 20, line 7: "..., reported concentrations ...": Please give the reference where they were reported.

**Reply:** The same misunderstanding, Thank you. We add the reference: Higueras et al., 2014 also at page 19, line 8.

23. page 20, line 23/24: "... exported from ... by strong katabatic winds.": Did you investigatethis? How is it proven? Or is it taken from another publication? Then it should becited, here.

**Reply:** Yes, thanks. We reported the publication "Angot et al., 2016a" in this issue where investigation on this as well as related discussion have been reported.Please, see page 19, line 25.

24. page 20, line 23: "Observations at DDU also highlighted ... ": The reader cannot followthis. Where can this be seen?

**Reply:** In this special Issue, Angot et al. discussed the observations at DDU. Please, see the reference "Angot et al., 2016a" reported at page 19, line 25.

25. page 21, lines 1/2: What about Celestun? According to Table 1, this is in the tropics,too.

**Reply:** Yes, Celestùn (Mexico) is in the tropics and as specified earlier in the manuscript, Celestùn is a site that start Hg measurements in 2012 but it performed Hg measurements only in that year because it was relocated with Sisal (Mexico) site that start measurements in 2013 till the end of the project. Therefore, because our discussion is primarily focused on 2013-2014 period, we considered the data from Sisal, ruling out Celestùn from the discussion.

26. page 21, line 11: What do you mean with "meteo-climatic conditions"?

**Reply:** Meteo-climatic factors and/or variables (i.e., temperature, rain amount, pressure, relative humidity, wind speed and direction as well as pressure systems etc.). KOD is a high altitude site (2333 m a.s.l.). Therefore different meteo-climatic conditions influence the long range transport of air masses to this site (Pirrone et al., 2016 this issue, in preparation).

27. page 21. line 12. How close is the thermometer plant?

**Reply:** It is at 2150 m. Please, see now at page 20, line 15. Thank you.

28. page 21, line 14: " India is the third largest hard coal producer ...": Concerning Hg emissions, it is more important how much is consumed.

**Reply:** Yes, we also agree with you on this. From the "*EIA – Independent Statistics and Analysis, U.S. Energy Information Administration*"(www.indexmundi.com) as well as within the report "Coal in India 2015" by the Australian Government - Department of Industry and Science, ISBN: 978-1-925092-63-9", (which has been now reported within the "Reference section") it has been a comparable increasing of the coal production and consumption in India from 1980 to 2014 yrs. The "EIA" in particular shows an increase of coal production by year from about 128 thousand tons in 1980 to 675.5 thousand tons in 2013 and a parallel increasing of coal consumption by year from 119 thousand tons in 1980 to 886 thousand tons in 2013.Detailed information have also been reported within the cited report above, therefore, in order to give more information on this, we have reported within this section of the manuscript the references of them. Thank you for your comment.Please, see the citation "Penney and Cronshaw, 2015" reported in the text at pag.20, line 18.

29. page 21, line 23: "... it is necessary for India ...": I think this necessary at other places in the world, too.

**Reply:**Yes, we agree with you, therefore, we highlighted your comment within the sentence as follow:

"…Therefore it is necessary for India as well as for the other places in the world where Hg measurements are yet lacking to generate continuous data, which can be used by scientists for modelling applications to improve emission inventories in order to prevent inaccurate assessments of Hg emission and deposition."….

Please, see page20, line 26 - 28. Thank you.

30. page 21, line 30/31:"SIS site was typically influenced by ...": Again you claim something that the reader is not able to follow. This leaves the impression that you speculate orthat it is published somewhere else and you do not give a reference.

**Reply:** yes, thank you. We reported the reference related to a scientific report by Sena et al., 2015 where this analysis has been performed. Please,see in the revised manuscript at page21, line 1-2.

31. page 22, line 11/12: Please explain how you analysed the influence of different airmasses on the concentrations measured at NIK.

**Reply:** yes, thank you very much for your comment. We analyzed the influence of air masses on the Hg concentrations measured at NIK using the backward trajectories by the Hybrid single-particle Lagrangian integrated trajectory model (HYSPLIT) available at the NOAA Air Resources Laboratory (Air Resources Laboratory 2010), to calculate 5 or 10-day long backward trajectories (Draxler and Rolph 2003). Please, see Figures SM1 and SM2 within the Supplementary Material the backward trajectory analysis performed for NIK site during specific periods with high GEM concentrations recorded in order to see potential influence of air masses crossing the site on the concentrations measured. Thank you once more.

32. page 22, line 34: Here you refer to the year 2012 while all other stations were onlyevaluated for 2013/2014.

**Reply:**Yes, we evaluated the two years (2013/2014) with a major data coverage(%) of the observations as pointed out earlier. In fact, our statistical evaluations/calculations are related to this periodfor all the ground-based sites taken into account within the GMOS network in order to harmonize the discussion and compare the results worldwide. However, this not exclude the possibility that somewhere we reported some comments or comparable data with other observations (as is the case for some results obtained during other studies above all at some locations where Hg measurements were not performed before like at Cape Verde). Thank you for your comment.

33. page 22, line 35, page 23, line 1: Again, the reader gets the impression that you analysed the meteorological conditions at the different stations but you do not explain howyou did this.

**Reply:**Thank you for your comments on the issue "meteorological conditions" related to some GMOS monitoring stations. I'd like to highlight that all data coming from the monitoring stations have been analyzed considering the ancillary data transmitted to the GMOS central database from all ground-based sites of the network and analyzing the different meteorological conditions and data of all stations discussed within the manuscript together with all involved scientific responsible and site managers of each station and reportedin the coauthors' list of the manuscript itself, therefore we didn't speculate anything, but, probably, we didn't make clear the discussion leaving out somewhere the reference of previously published works by some coauthors on GMOS data related to, for example, to a specific station during a specific period of the project.In this case, following your comment related to page 22 and 23, we insert as reference the PhD thesis of Luis Silva Mendes Neves in which is specifically discussed this issue on Cape Verde data. Thank you very much for your input. Please, see at page 22, line 21.

34. page 23, line 18/19: Over which years has the inter hemispherical gradient been constant? You analyse only two year in this paper. Or do you refer to other investigations?

**Reply:**The overall goal of the manuscript was to present the monitoring data obtained worldwide in the framework of the GMOS network reporting continuous Hg measurements also in place of the world where Hg measurements were lacking before the establishment of the GMOS, such as some sites located between the Tropics, and particularly in the Southern Hemisphere in order to provide a consistent set of high quality data useful for modeling application on regional and global scale to better understand Hg chemistry and processes worldwide. In several places of the Southern Hemisphere Hg observations were performed during ad-hoc measurements campaigns and/or oceanographic campaigns (also performed during the GMOS project as part of the project itself within the WP4 leaded by Prof. Milena Horvat, please, see at the web page of the GMOS project: www.gmos.eu ). With GMOS continuous Hg measurements were carried out and are to date ongoing, therefore, one of the goal of our analysis is to confirm the inter-hemispherical gradient of Hg concentrations observed from the Northern Hemisphere to the Southern Hemisphere as argued in previous works published in the literature and based both on ad-hoc Hg measurements temporary limited in several places of the world and across existing regional Hg network. Therefore, within the "Conclusion" section of our manuscript we highlighted that:…"The inter-hemispherical gradient with higher GEM concentrations in the Northern Hemisphere has remained nearly constant over the years, and confirmed by the observations carried out in the Southern Hemisphere and other locations where before GMOS Hg measurements were lacking or absent."…Thank you for your comment.Please, see Conclusions now at page 23.

35. page 23, line 20: You mention cruises but there is no data from cruises presented here.

**Reply:**  yes, we referred to cruise campaigns previously carried out. Please, see our reply before. Thank you.

**Technical corrections:**

1. affiliation 14: Sweden

**Reply:**  Corrected, Thank you.

2. Write all units correctly, e. g. on page 2, line 24: convert ng m$^{-3}$ into ngm$^{-3}$. This appears at several places in the entire document.

**Reply:**  Corrected the unit on page 2, now line 26  as well as in other places of the document where have been highlighted. Thank you.

3. Improve the way citations are shown, by removing inner brackets when two or more referencesare given, e.g. page 2, line 23: (Lindberg et al., 2007; Sprovieri et al., 2010b).This appears at several places in the entire document.

**Reply:**  Correctedin this page as well as in other places of the entire document. Please, see through the revised manuscript.Thank you very much.

4. Explain TGM earlier in the document. It appears first on page 3, line 32 but is explainedlater.

**Reply:**  The sentence: …."Master Stations perform speciated Hg measurements and collect precipitation samples for Hg analysis whereas the Secondary Stations perform only TGM/GEM measurements."….

has been replaced with ….

…."Master Stations perform speciated Hg measurements and collect precipitation samples for Hg analysis whereas the Secondary Stations perform only Total Gaseous Mercury (TGM)/GEM measurements and precipitation samples as well."….Please, see at page 4, line 10.Thank you.

5. Take care about words starting with capitals. E.g. on page 4, line 2, I propose to write"GMOS external partners". There are several other places where the use of capitalsshould be re-considered.

**Reply:** Corrected on page 4, line 14 and elsewhere according to. Thank you.

6. Explain PBM2.5 on page 4.

**Reply:** Done. Please, see pag. 6, line 7.Thank you.

7. Give more explanations about what is displayed in Table 1 and Table 2 in the respective captions.

**Reply:**The Caption related to Table 1has been rewritten according to previous comments reported above as well as the caption related to Table 2 which is now "Table SM1" inserted within the "Supplementary Material"doc.

8. The graphs in Fig. 1 and Fig. 2 are too small and therefore hard to read.

**Reply**: We agree. Both Fig.1 and Fig. 2 have been enlarged according to. Thank you. Please, see now at page 9.

9. page 11, line 4: "latitude" instead of "longitude".

**Reply**:  Corrected, Thank you.Please, see now at page 11, line 10.

10. Fig. 4 and Fig. 5: If one has a title the other should also have one. They could be a bit smaller or combined into one figure with two panels. It is not clear what "raw data" in the caption of Figure 4 really means.

**Reply:**  Figure 4 has been completely replaced according to your comment within the "major comments" section and a little bit reduced in size following your input. Figure 5 has been deleted as it was redundant as also pointed out by the referee. Regarding the "raw data" means, please, see our reply to your comment on it within the section "Major Comments", even if this definition has been deleted within the caption, now  rewritten according to the new Figure 4. Thank you.

11. Table 5 looks misplaced, it is first mentioned 5 pages later. The number 0.0.365 is misspelled.

**Reply:**  We agree, Table 5, now Table 3, has been replaced close to the page where it is mentioned according to. The number has been corrected as well. Thank you. Please, see now at page 17.

12. Figure and table captions should end with a full stop.

**Reply:**  Thank you, done for all Figure and Table captions.

13. Abstract, page 1, line2: under stand

**Reply:** Corrected, Thank you.Please, see now at page 2, line 2.

14. page 2, line 30: Wängberg

**Reply:** Corrected, Thank you. Please, see now at page 2, line 32.

15. page 3, line 14: and a QA/QC ...

**Reply:** Corrected, Thank you. Please, see now at page 3, line 20.

16. page 4, line 15: ... at the French site Dumont d'Urville.

**Reply:** Corrected according to. Thank you. Please, see now at the end of page 5.

17. page 4, line 29: ... within the GMOS network.

**Reply:** Corrected, Thank you.Please, see now at page 6, line 15.

18. page 6, line 4: This is also in line with a study recently published by Slemr et al. (2015) ...

**Reply:** Corrected, Thank you. Please, see now at page 6, line 25.

19. page 6, line 8: located at the coastline

**Reply:** Corrected, Thank you. Please, see now at page 6, line 29.

20. page 9, line 20: raw data

**Reply:** Corrected, Thank you. Please, see now at page 8, line 21.

21. page 13, Table 5: 0.365. The table is misplaced because it is mentioned much later inthe text (page 16).

**Reply:**The value highlighted has been corrected and the Table 5(now Table 3) has been replaced as well following your suggestion. Please**,** see now at page 17. Thank you.

22. page 15, line 12/13: misplaced brackets for the references.

**Reply:** Corrected according to. Thank you.Please, see now at page 12, line 23 and 24.

23. page 15, line 15: Which station is STN?

**Reply:** I'm sorry, STN wasthe old code of Station Nord which was changed with VRS (Greenland), upon request of the site manager. I'm sorry it was lost with the old codein this part of the manuscript but now has been replaced with VRS. Please, see now at page 14, line 1. Thank you.

24. page 17, line 3/4: ... the three distributions come from the same ...

**Reply:** This part of the manuscript has been revised and rewritten according to the major comment reported above. Please, see now at page 16, line 7. Thank you.

25. page 17, line 7: Table 5 is on page 13, which is not close enough to the page where it's mentioned.

**Reply:** We agree, Table 5 which, with the relocation of the other tables in the supplementary material doc, became now Table 3, has been replaced close to the page where it is mentioned according to. Please, see the revised manuscript at page 17. Thank you

26. page 17, line 23: ... during the years 2013 ...

**Reply**: Corrected, Thank you.Please, see the revised manuscript at page 17, line 7.

27. page 17, line 24: I think this has to be GOM (not GEM).

**Reply**: Corrected, Thank you.Please, see the revised manuscript at page 17, line 8.

28. page 18, line 7: ... at the edge of the north-eastern part ...

**Reply**: Corrected, Thank you.Please, see the revised manuscript at page 18, line 26.

29. page 18, line 9: Hg sources

**Reply**: Corrected, Thank you.Please, see the revised manuscript at page 18, line 1.

30. page 20, line 17: in the same period

**Reply**: Corrected, Thank you.Please, see the revised manuscript at page 19, line 18.

31. page 20, line 30: in the Tropics

**Reply**:Corrected, Thank you.Please, see the revised manuscript at page 19, line 32.

[revised manuscript text omitted]